# Sterol methyltransferases in uncultured bacteria complicate eukaryotic biomarker interpretations

Malory O. Brown [1], Babatunde O. Olagunju[2], José-Luis Giner [2] & Paula V. Welander [1]✉

Sterane molecular fossils are broadly interpreted as eukaryotic biomarkers, although diverse bacteria also produce sterols. Steranes with side-chain methylations can act as more specific biomarkers if their sterol precursors are limited to particular eukaryotes and are absent in bacteria. One such sterane, 24-isopropylcholestane, has been attributed to demosponges and potentially represents the earliest evidence for animals on Earth, but enzymes that methylate sterols to give the 24-isopropyl side-chain remain undiscovered. Here, we show that sterol methyltransferases from both sponges and yet-uncultured bacteria function in vitro and identify three methyltransferases from symbiotic bacteria each capable of sequential methylations resulting in the 24-isopropyl sterol side-chain. We demonstrate that bacteria have the genomic capacity to synthesize side-chain alkylated sterols, and that bacterial symbionts may contribute to 24-isopropyl sterol biosynthesis in demosponges. Together, our results suggest bacteria should not be dismissed as potential contributing sources of side-chain alkylated sterane biomarkers in the rock record.

Sterols are polycyclic triterpenoid lipids required by most eukaryotes for many critical cellular functions, such as maintaining membrane integrity, phagocytosis, stress tolerance, embryogenesis, and cellular signaling[1–6]. In addition, sterols are quite recalcitrant and are preserved as sterane hydrocarbons in sedimentary rocks up to 1.6 billion years old[7]. Given their persistence in the geologic record and wide distribution amongst extant eukaryotes, steranes are broadly interpreted as biomarkers for ancient eukaryotic life[8,9], although diverse bacteria are also known to produce sterols[10]. Sterols with particular modifications to the side-chain, such as methylations at the C-24 and C-26 positions, have not been reported in bacteria and remain unique to certain eukaryotic lineages. Side-chain alkylated steranes, such as ergostane and stigmastane, can therefore function as more specific biomarkers[11].

According to the sponge biomarker hypothesis, two side-chain alkylated sterane biomarkers, 24-isopropylcholestane (24-ipc; Fig. 1a)

and 26-methylstigmastane (26-mes), are indicative of marine sponges of the Demospongiae class[12,13]. The co-occurrence of 24-ipc and 26-mes in the late Neoproterozoic (660-540 Ma)[12,13], preceding the earliest sponge macrofossils by ~100 million years[14], may represent the first evidence of animals in the geologic record. However, debate over this interpretation continues for several reasons[14–22] including the potential for alternative sources. Demosponges are the only extant organisms known to contain sterols with the 24-ipc and 26-mes side-chain structures as their major sterols[13,23–27], but pelagophyte algae, which produce 24-n-propylcholesterol as their major sterol, also contain 24-isopropyl sterols[28–31]. Trace amounts of 24-ipc and 26-mes were also reported in hydrogenated lipid extracts from rhizarian protists[22], although attempts to replicate these results were not successful[20]. Subsequent studies demonstrated 24-ipc and 26-mes form abiotically from certain $C_{29}$ sterols during laboratory pyrolysis experiments under conditions that may mimic diagenesis[18,21]. These studies suggest that

[1]Department of Earth System Science, Stanford University, Stanford, CA 94305, USA. [2]Department of Chemistry, State University of New York-Environmental Science and Forestry, Syracuse, NY 13210, USA. ✉e-mail: welander@stanford.edu

the previously observed Rhizarian 24-ipc and 26-mes were generated through laboratory pyrolysis rather than by in vivo synthesis, and that Neoproterozoic 24-ipc and 26-mes may have formed abiotically from algal sterols. Further, microbial symbionts, including bacteria from diverse phyla, can constitute up to 40% of sponge biomass[32], and recent analyses have identified bacterial sterol biosynthesis gene clusters in several demosponge metagenomes[33,34]. Bacterial symbionts may therefore contribute to 24-isopropyl and 26-methyl sterol biosynthesis in sponges, although bacterial enzymes that methylate the sterol side-chain have not been previously identified.

One approach to constrain uncertainties in the sponge biomarker hypothesis is to identify and characterize the enzymes responsible for these distinct alkylations at C-24 and C-26. Sterol 24-C-methyltransferases (SMTs) alkylate the C-24 position at an isolated double bond in an *S*-adenosylmethionine (SAM) dependent mechanism[35] and have been characterized extensively in fungi and plants. Bioinformatic analyses performed to understand how sponges could be synthesizing 24-propyl sterols suggested that the number of SMT copies in a eukaryotic genome corresponds to the number of carbons added at C-24 in all analyzed lineages except sponges, which contain one less SMT in their genome or transcriptome than expected based on their sterol profile[36]. This prompted the hypotheses that at least two SMT copies in a genome are required to produce 24-propyl sterols, and that sponges each encode bifunctional SMTs that can perform multiple methylations at C-24. However, no experimental evidence verifying this activity by any sponge-derived SMT currently exists, and SMTs that produce 24-propyl sterols have not been identified in any extant organism.

Here, we characterize a variety of putative SMTs from sponges and bacteria to better understand the biochemical requirements for generating side-chain alkylated sterols, including 24-isopropyl sterols. We demonstrate that sponge SMT homologs are bona fide SMTs that can perform multiple rounds of methylation to produce sterols methylated and ethylated at the C-24 position. In addition, we show that SMTs from yet-uncultured bacteria, including demosponge symbionts, are capable of alkylating sterols at the C-24 position. We identify three symbiotic bacterial SMTs each capable of performing all three of the methylation steps required to produce the 24-isopropyl side-chain. Taken together, our results suggest yet-uncultured bacteria have the genomic capacity to synthesize side-chain alkylated sterols and demonstrate the utility of understanding the biochemistry of lipid biosynthesis to allow for more robust biomarker interpretations.

## Results

### Sponge SMTs are functional

Our first goal was to express putative sponge SMTs and verify methylation activity in vitro. Given the current lack of publicly available sequencing data from sponges known to contain 24-propyl sterols, the function of SMTs from these species could not be verified in this study. We therefore chose eight SMT homologs identified in publicly available genomes and transcriptomes from eight sponges of the Demospongiae, Homoscleromorpha, and Calcarea classes, each of which have been shown to contain 24-ethyl sterols at the species or genus level (Supplementary Table 1). In vitro reactions with cell lysates generated from *Escherichia coli* strains overexpressing each SMT were performed with two potential substrates, desmosterol and 24-methylenecholesterol, which have been previously shown to be suitable substrates for side-chain methylation in sponges[37], pelagophyte algae[29], and annelid worms[38]. Sterol products were analyzed using gas chromatography-mass spectrometry (GC-MS). All eight of the sponge SMTs methylated desmosterol to produce 24-methyl sterols including 24-methylenecholesterol and (epi)codisterol (Figs. 1c, 2). Six SMTs also methylated 24-methylenecholesterol to 24-ethyl sterols including fucosterol, isofucosterol, and (epi)clerosterol (Figs. 1d, 2). Each

sponge SMT capable of producing 24-ethyl sterols from 24-methylenecholesterol could also bifunctionally methylate desmosterol. None of the sponge SMTs produced 24-propyl sterols, consistent with the sterols previously observed in these species (Supplementary Table 1).

### Three SMTs from symbiotic bacteria produce 24-isopropyl sterols

Our analyses of sponge symbiont metagenomes revealed several SMT homologs, many of which occurred in gene clusters with homologs of other known sterol biosynthesis genes including the bacterial C-4 demethylases *sdmA* and *sdmB*[39,40] (Fig. 3, Supplementary Data 1). We therefore hypothesized bacterial symbionts may contribute to side-chain alkylation in sponges. However, no bacterial enzymes, from either free-living or host-associated sources, have been experimentally shown to alkylate the sterol side-chain. We chose 10 putative bacterial SMTs identified in metagenomes from four demosponge species known to contain side-chain alkylated sterols (Supplementary Table 1) for analysis. Eight symbiont SMTs methylated desmosterol to 24-methyl sterols including (epi)codisterol, 24-methylenecholesterol, and 24-methyldesmosterol (Figs. 1c, 2). Three of these SMTs also methylated 24-methylenecholesterol to 24-ethyl sterols including fucosterol, isofucosterol, and (epi)clerosterol (Figs. 1d, 2). As with the bifunctional sponge SMTs, each bacterial symbiont SMT that produced 24-ethyl sterols from 24-methylenecholesterol also methylated desmosterol. Two SMTs identified in a metagenome from *Aplysina aerophoba*, a demosponge known to contain trace 24-ipc after catalytic hydropyrolysis[13], also methylated 24-methylenecholesterol to two distinct $C_{30}$ sterols (Figs. 1, 2). The first symbiont SMT produced 24S−24-isopropylcholest-5,25-dienol while the second produced 24-isopropylcholest-5,24-dienol. Both propylating SMTs were capable of all three methylation steps required to synthesize the 24-ipc side-chain via a single protein. One propylating SMT was identified in a sterol biosynthesis gene cluster suggesting a yet-unknown bacterial sponge symbiont has the genomic capacity to synthesize 24-isopropyl sterols de novo (Fig. 3, Supplementary Data 1).

Identification of bacterial sponge symbiont SMTs capable of side-chain alkylation led us to question if bacteria from environments outside a sponge host could also produce side-chain alkylated sterols. Through additional searches, we identified SMT homologs in a variety of environmental metagenomes and metagenome-assembled genomes (MAGs). We tested 14 of these SMT homologs from marine, freshwater, thermal spring, and coral reef environments, five of which occur in sterol biosynthesis gene clusters (Fig. 3, Supplementary Data 1). Twelve of these SMTs alkylated the sterol side-chain to produce 24-methyl and/or 24-ethyl sterols (Figs. 1, 2). Each environmental SMT that methylated 24-methylenecholesterol also methylated desmosterol. Two of the functional SMTs were identified in a sterol biosynthesis gene cluster from a *Sandaracinus* sp. MAG assembled from a pelagic North Atlantic Ocean metagenome[41] (Fig. 3). The MAG also contains homologs for demethylation at C-4 and C-14 (Supplementary Data 1), suggesting a marine myxobacterium has the genomic potential to produce side-chain alkylated regular sterols de novo. One SMT, identified in a drinking water filter MAG assigned to the bacterial phylum Chlamydiae, produced both 24S−24-isopropylcholest-5,25-dienol and 24-isopropylcholest-5,24-dienol (Fig. 1).

The identification of functional metagenomic SMTs likely derived from bacteria, including three that produce 24-isopropyl sterols, is significant given that no bacterial proteins had previously been shown to alkylate the sterol side chain. To confirm that these SMTs are truly of bacterial origin, we performed phylogenetic analyses of the 24 metagenomic SMTs we tested. In our maximum-likelihood tree of SMT proteins the majority of the metagenomic SMTs are distinct and more closely related to each other than to the SMTs of eukaryotes known to produce side-chain alkylated sterols (Fig. 4a). Further, 23 of the 24

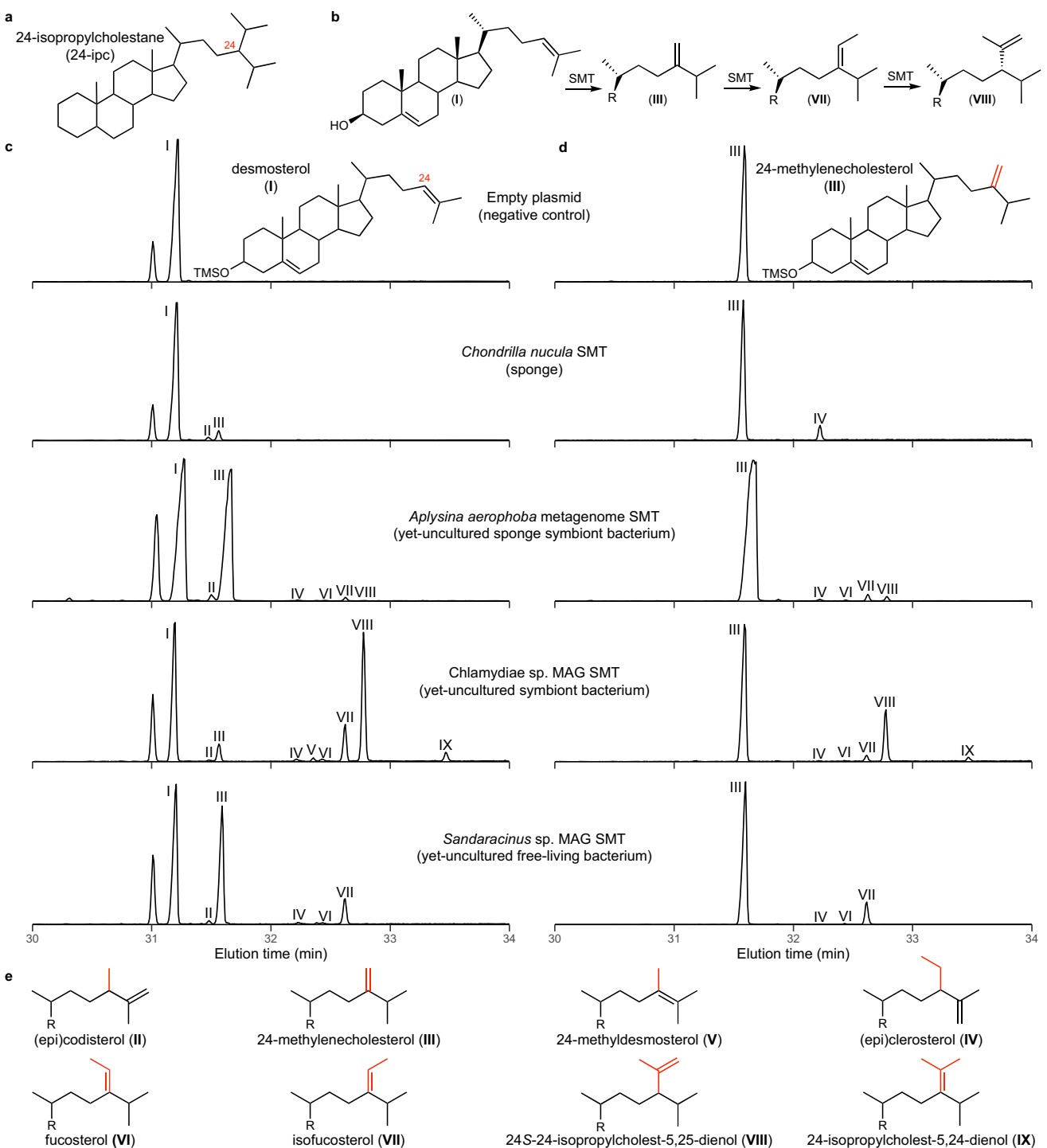

**Fig. 1 | Sterol methyltransferases from sponges and yet-uncultured bacteria perform multiple side-chain methylations in vitro. a** Structure of the sponge biomarker 24-isopropylcholestane. **b** Predominant pathway to 24-isopropyl sterols by bacterial sterol methyltransferases (SMTs). **c** Representative extracted ion chromatograms (m/z 456, 470, 484, 498) of total lipid extracts from in vitro reactions performed with desmosterol and E. coli lysates containing an empty plasmid, an SMT identified in a transcriptome of the demosponge Chondrilla nucula, a bacterial SMT identified in a metagenome from the demosponge Aplysina aerophoba, a bacterial SMT identified in a freshwater metagenome-assembled genome (MAG) assigned to the phylum Chlamydiae, and a myxobacterial SMT identified in a marine MAG assigned to the genus Sandaracinus. **d** Representative extracted ion chromatograms (m/z 456, 470, 484, 498) of total lipid extracts from in vitro reactions performed with the same lysates but with 24-methylenecholesterol as the substrate. **e** Side-chain structures of sterols identified in **c** and **d**. All lipids were derivatized to trimethylsilyls prior to GC-MS analysis. The raw data used to generate this figure are provided in Supplementary Data 2. Mass spectra of identified sterols are shown in Supplementary Fig. 3.

putative bacterial SMTs, including those from sponge metagenomes, form two clades separate from the sponge SMT clade. It is therefore unlikely that these metagenomic SMTs were acquired from sponges via horizontal gene transfer. However, the observed bootstrap values were not robust enough to draw any additional evolutionary conclusions, nor to eliminate any remaining uncertainty of the bacterial origin of these SMTs. We therefore used the recently developed classifier Whokaryote, which predicts whether a metagenomic contig is likely

| | C28 | | | C29 | | | C30 | |
|---|---|---|---|---|---|---|---|---|
| | (II) | (III) | (V) | (IV)* | (VI) | (VII) | (VIII) | (IX) |
| **DEMOSPONGIAE** | | | | | | | | |
| *Amphimedon queenslandica* | | + | | | + | | | |
| *Chondrilla nucula* | + | + | | + | | | | |
| *Sarcotragus fasciculatus* | + | + | | | + | | | |
| *Haliclona amboinensis* | + | + | | + | + | + | | |
| *Haliclona tubifera* | + | + | | + | | | | |
| **HOMOSCLEROMORPHA** | | | | | | | | |
| *Oscarella pearsei* | + | + | | + | + | + | | |
| *Oscarella carmela* | + | + | | | | | | |
| **CALCAREA** | | | | | | | | |
| *Sycon ciliatum* | + | + | | | | | | |
| **DEMOSPONGE METAGENOME** | | | | | | | | |
| ***Aplysina aerophoba* JGIcombinedJ30088_10000835 1 of 2** | + | + | | + | + | + | + | |
| ***Aplysina aerophoba* JGIcombinedJ30088_10000835 2 of 2** | | | | | | | | |
| *Aplysina aerophoba* Ga0209021_1000146 | + | + | + | | + | + | | + |
| ***Petrosia ficiformis* LXNJ01014139** | | + | | | | | | |
| ***Petrosia ficiformis* LXNJ01003202** | | + | | | | | | |
| *Petrosia ficiformis* LXNJ01000389 | + | + | + | | + | | | |
| ***Petrosia ficiformis* LXNJ01004423** | | | | | | | | |
| ***Sarcotragus foetidus*** | | + | | | | | | |
| **DEMOSPONGE METAGENOME-ASSEMBLED GENOME** | | | | | | | | |
| ***Theonella swinhoei* associated Chromatiales sp. 1 of 2** | | + | | | | | | |
| *Theonella swinhoei* associated Chromatiales sp. 2 of 2 | | | | | | | | |
| **OTHER METAGENOME** | | | | | | | | |
| **Marine sediment, Gulf of Thailand** | | + | | | | | | |
| Marine sediment, Helgoland, North Sea | | + | | | | | | |
| Western Arctic Ocean | + | + | | + | + | + | | |
| Freshwater, Lake Fryxell, Antarctica | + | + | | | + | + | | |
| Lake sediment, Walker Lake, Nevada | | + | | | | | | |
| Hot spring, Beatty, Nevada | | | | | | | | |
| Hot spring sediment, Dewer Creek, BC | | + | | | | | | |
| **OTHER METAGENOME-ASSEMBLED GENOME** | | | | | | | | |
| Chlamydiae sp., drinking water 1 of 2** | + | + | + | + | + | + | + | + |
| Chlamydiae sp., drinking water 2 of 2 | + | | + | | | | | |
| ***Nitrospira* sp., coral reef 1 of 2** | + | + | | + | | | | |
| ***Nitrospira* sp., coral reef 2 of 2** | | | | | | | | |
| ***Sandaracinus* sp., marine 1 of 2** | + | + | | | | | | |
| ***Sandaracinus* sp., marine 2 of 2** | + | + | | + | + | + | | |
| Spirochaetes sp., soil | | + | | | | | | |

**Fig. 2 | Sterols identified as products of sponge and bacterial sterol methyl-transferase in vitro reactions.** Sterols were detected using GC-MS. Products of in vitro reactions performed with both desmosterol and 24-methylenecholesterol as substrates are included. C28 sterols include (epi)codisterol (**II**), 24-methylenecholesterol (**III**), and 24-methyldesmosterol (**V**). C29 sterols include (epi) clerosterol (**IV**), fucosterol (**VI**), and isofucosterol (**VII**). C30 sterols include 24*S*−24-isopropylcholesta-5,25-dienol (**VIII**) and 24-isopropylcholesta-5,24-dienol (**IX**). Metagenomic SMTs identified in gene clusters with at least two other sterol biosynthesis homologs are bolded. *NMR confirmed epiclerosterol and not clerosterol as products of the "*Aplysina aerophoba* JGIcombinedJ30088_10000835 1 of 2" and "Chlamydiae sp., drinking water 1 of 2" sterol methyltransferases. **24-iso-propylcholesta-5,23-dienol was also detected as a product of the "Chlamydiae sp., drinking water 1 of 2" sterol methyltransferase by NMR, but was not detected by GC-MS.

eukaryotic or prokaryotic based on intergenic distance, gene density, gene length, and k-mer frequencies. Whokaryote predicts all 19 of the contigs containing the metagenomic SMTs tested here to be of prokaryotic origin (Supplementary Table 2).

The genomic context of the SMTs we tested provides additional evidence of their bacterial origin. Of the 24 metagenomic SMTs, 13 were in gene clusters or MAGs with a homolog of oxidosqualene cyclase (OSC), a key enzyme in sterol biosynthesis responsible for cyclizing oxidosqualene to lanosterol[10] (Fig. 3, Supplementary Data 1). OSCs cluster into robust phylogenetic clades with high bootstrap support[42]. Thus, phylogenetic analyses of OSCs associated with the SMTs we tested could confirm their bacterial origin. In our maximum-likelihood tree of OSC proteins (Fig. 4b), we see that seven of the OSCs associated with the SMTs analyzed here, including four from demosponge metagenomes, fall into the Group 1 clade which contains only bacterial OSCs. The remaining 2 OSCs, from soil and hot spring sediment metagenomes, fall into bacterial Group 2. The position of these

OSCs provides additional evidence that the metagenomic SMTs we tested originate from a bacterial source.

## Side-chain propylation mechanism

It was surprising to observe that one SMT was capable of propylating the sterol side-chain on its own as it has been hypothesized that at least two SMTs would be required for this reaction[36]. To better understand this unexpected biosynthetic pathway, we performed additional in vitro reactions with two of the bacterial propylating SMTs using [13]C-labeled sterol substrates and analyzed the resulting products by nuclear magnetic resonance (NMR; Supplementary Table 3, Supplementary Figs. 2, 4–7). In experiments with [28-13C] 24-methylenecholesterol as the substrate of the *A. aerophoba* sponge symbiont SMT, the major C29 products were confirmed as iso-fucosterol and fucosterol present in a 7:1 ratio. Trace epiclerosterol, but not clerosterol, was also detected. The product of triple side-chain methylation was confirmed as 24*S*−24-isopropylcholesta-5,25-dienol.

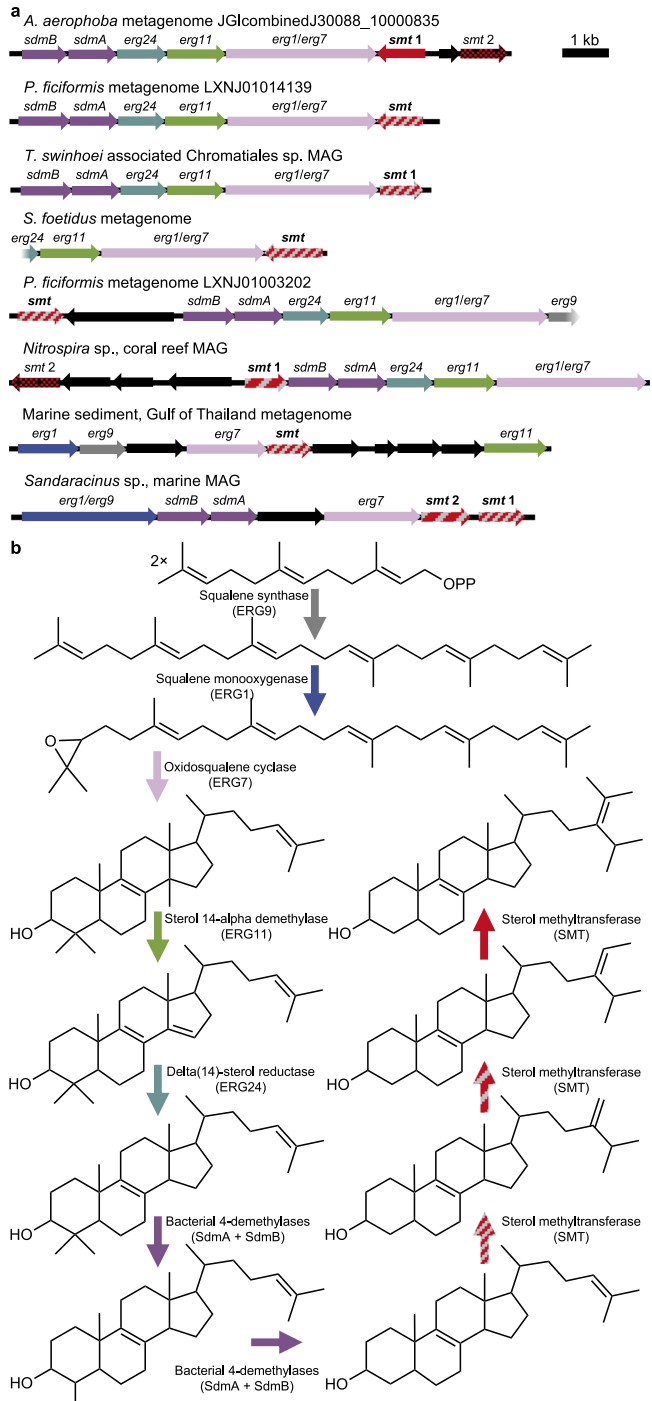

**Fig. 3 | Gene clusters suggest de novo side-chain alkylated sterol biosynthesis in the bacterial domain. a** Gene clusters identified in metagenomes and metagenome-assembled genomes (MAGs) containing three or more sterol biosynthesis homologs and including functional sterol methyltransferases (bold). Genes are named for their *Saccharomyces cerevisiae* (*erg*) or *Methylococcus capsulatus* Bath (*sdmAB*) homologs. Dotted sterol methyltranferases did not methylate desmosterol or 24-methylenecholesterol in vitro. Striped patterns correspond to the number of methylations performed by each sterol methyltransferase in vitro. Gradients indicate partial sequences. **b** Schematic of a hypothetical bacterial side-chain alkylated sterol biosynthesis pathway. Arrow colors at each step correspond to the homologs identified in **a**. Detailed information on the source of each metagenomic gene cluster is available in Supplementary Table 4. Homology for each gene is described in Supplementary Data 1.

The label was found exclusively at C-25, and the 24 *R* isomer was absent (Supplementary Fig. 1a). Fucosterol was not a substrate for the *A. aerophoba* symbiont SMT, but [29-$^{13}$C] isofucosterol yielded 24*S*−24-isopropylcholesta-5,25-dienol with the label distributed in a 5:1 label ratio between C-27 and C-26 (Supplementary Fig. 1b). This outcome shows interconversion of the C-28 and C-25 cations by hydride-shifts is not involved in the biosynthesis of this compound, and deprotonation of the initially formed cation is unspecific.

The same set of reactions with the propylating Chlamydiae MAG SMT gave similar results. Fucosterol was not a substrate and epiclerosterol, but not clerosterol, was detected. However, minor differences were measured between these two bacterial SMTs. In experiments with [29-$^{13}$C] isofucosterol and the Chlamydiae MAG SMT, the ratio of isofucosterol and fucosterol was 30:1, and the ratio of label at C-27/C-26 of 24*S*−24-isopropylcholesta-5,25-dienol was 6:1. In addition, the C$_{30}$ Δ23 and Δ24 isomers, 24-isopropylcholesta-5,23-dienol and 24-isopropylcholesta-5,24-dienol, were also detected. The label was detected in the Z-vinylic methyl of 24-isopropylcholesta-5,24-dienol and in one of the two methyls of the Z-isopropyl group of 24-isopropylcholesta-5,23-dienol. These results provide evidence for one hydride-shift to generate the C-24 cation. The ratio of 24*S*−24-isopropylcholesta-5,25-dienol and its Δ$^{24}$ and Δ$^{23}$ isomers was estimated to be 20:4:1, respectively. Though minor mechanistic differences between these two propylating SMTs exist, the predominant pathway to 24-isopropyl sterols observed here proceeds from desmosterol to 24-methylenecholesterol to isofucosterol to 24*S*−24-isopropylcholest-5,25-dienol (Fig. 1b).

## Discussion

The results presented here experimentally verify that sponges do indeed encode functional sterol methyltransferases capable of sequentially methylating desmosterol to 24-methylenecholesterol to 24-ethyl sterols, similarly to the bifunctional SMTs recently identified in annelid worms[38]. However, as in Annelida[43], this activity was inconsistent among sponges. All five of the demosponge SMTs we tested sequentially methylated sterols. However, this activity was not observed by the SMTs from *Oscarella carmela* (Homoscleromorpha) and *Sycon ciliatum* (Calcarea), contrary to a previous hypothesis that all sponges encode bifunctional SMTs[36]. Sterol profiles from *O. carmela* and *S. ciliatum* have not been reported, but 24-ethyl sterols have been identified in other members of *Oscarella* and *Sycon*[44,45]. Sponge-derived bifunctional SMTs that produce 24-ethyl sterols may therefore be restricted to certain species of Homoscleromorpha and Calcarea, although we recognize these SMTs may require a different substrate than we provided in our assays in order to sequentially methylate sterols. Substrate specificity could also explain why no sponge SMT produced 24-isopropyl sterols in our experiments, although this result was unsurprising given that we did not test SMTs from sponges known to contain propylated sterols due to the lack of available genomic data. Future sequencing of genomes or transcriptomes from demosponges known to contain sterols with the 24-ipc carbon skeleton may reveal sponge-derived SMTs sufficient for the biosynthesis of 24-propyl sterols.

While we were unable to demonstrate the production of propylated sterols through the expression of sponge SMTs, we did show that bacterial symbionts have the genomic capacity to synthesize these lipids and may therefore contribute to the 24-isopropyl sterol pool in demosponges. Previous studies have provided evidence of de novo sterol biosynthesis in demosponges[37,46,47], but the experiments were performed prior to the development of 16S community sequencing using cell-free sponge extracts or whole sponge tissue from which the presence of bacterial symbiont-derived enzymes cannot be excluded. Without confirmation that the bacterial symbionts in these samples did not contain functional SMTs, the possibility remains that bacterial symbionts contribute to sterol biosynthesis in demosponges. Further,

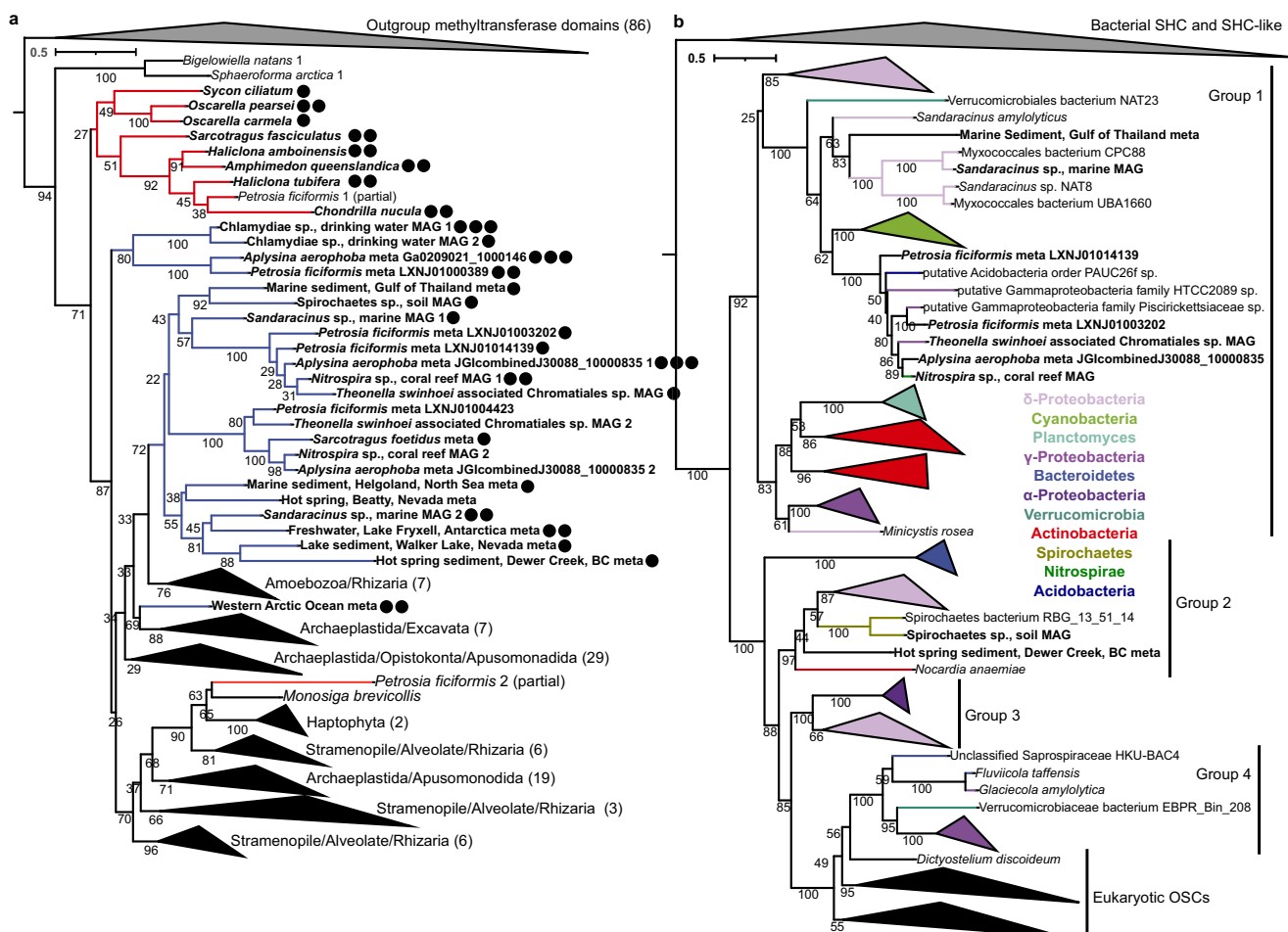

**Fig. 4 | Functional metagenomic SMTs are bacterial. a** Maximum likelihood tree generated from a concatenated alignment of the conserved methyltransferase and C-terminal domains of sterol methyltransferase (SMT) proteins. Metagenomic SMTs are in blue, sponge SMTs are in red, and other eukaryotic SMTs are in black. Bolded labels indicate SMTs tested in this study, and the number of circles corresponds to the number of carbons the SMT added to desmosterol at the C-24 position in vitro. The numbers in parentheses indicate the number of proteins in collapsed clades. **b** Maximum likelihood tree of oxidosqualene cyclases (OSCs). OSC groups are assigned as in ref. 42. Bolded labels indicate OSCs in metagenome gene clusters or metagenome-assembled genomes with SMTs tested in this study.

holo-transcriptomic sequencing of the demosponge *Vaceletia* sp. found that some sterol biosynthesis transcripts were sponge-derived, while others, including those for squalene synthase, squalene monooxygenase, and oxidosqualene cyclase, were derived from bacteria[48]. Both sponge and bacterial symbiont proteins may therefore be required to produce sterols in demosponges, similarly to what has been observed in certain deep-sea mussels[49,50]. However, the metagenomic gene clusters containing functional SMTs from *A. aerophoba* symbionts identified here, along with steroid gene clusters containing SMT homologs previously identified in bacterial MAGs associated with demosponges of the *Ircinia* genus[33], suggest bacterial symbionts may produce 24-isopropyl sterols de novo without demosponge input. Aplysterol, a 26-methyl sterol, is the major sterol of *A. aerophoba*, and trace 24-*n*-propyl sterols have also been reported in this demosponge[51]. While trace 24-ipc was reported after catalytic hydropyrolysis of *A. aerophoba* biomass, 24-isopropyl sterols have not been reported as components of *A. aerophoba*'s native sterols. It is therefore unlikely that bacterial symbionts produce significant amounts of 24-isopropyl sterols in this sponge. However, bacterial sterol biosynthesis gene clusters are enriched in demosponges compared to pelagic marine environments[33], and it remains to be seen if bacteria produce 24-isopropyl sterols in other demosponges. Our work highlights the need for robust sequencing of both the sponge host and its microbiome coupled to sterol analysis in demosponge species that contain

24-isopropyls as their major sterols. These coupled genomic and lipidomic analyses from relevant sponges would help confirm whether sponges, their bacterial symbionts, or some combination of both are potential sources of 24-ipc biomarkers.

The production of 24-isopropyl sterols by an SMT identified in a freshwater Chlamydiae MAG was surprising given that this bacterium is unlikely to produce sterols de novo due to the absence of other sterol biosynthesis homologs in the MAG. All characterized members of Chlamydiae are obligate endosymbionts of eukaryotes, and none have been shown to produce sterols. However, some Chlamydiae are known to recruit sterols from their eukaryotic host into their membranes[52]. Further, homologs of sterol reductases have been identified in members of this phylum, including in demosponge-associated MAGs[53]. Chlamydiae are found in both freshwater[54,55] and marine sponges[56–58], and a sterol identified in the demosponge *Callyspongia implexa* was shown to have antichlamydial activity[59]. The potential roles of chlamydial sterol modification during pathogenesis and symbiosis therefore warrant further investigation, and it remains to be seen if Chlamydiae methylate host-acquired sterols to give the 24-isopropyl side-chain in sponges and other eukaryotic hosts.

All three propylating SMTs we identified originated from bacterial symbionts associated with eukaryotic hosts, and SMTs from free-living bacteria capable of this reaction have yet to be discovered. However, we did identify SMTs capable of producing 24-methyl and 24-ethyl

sterols that likely originated in free-living bacteria. The two functional SMTs present in a sterol biosynthesis gene cluster from a marine *Sandaracinus* sp. MAG are particularly significant given that myxobacteria are rare in host-associated environments[60], and that several myxobacteria, including a *Sandaracinus* species, are known to produce sterols[10]. Further, a recent study demonstrated de novo cholesterol biosynthesis in the marine myxobacterium *Enhygromyxa salina* and identified bacterial *sdmB* homologs capable of complete demethylation at C-4[40]. Given that cultured marine myxobacteria are known to produce complex sterols fully demethylated at C-4 and C-14, and that a yet-uncultured marine *Sandaracinus* species encodes functional SMTs, we hypothesize that marine myxobacteria are the most promising candidates for de novo side-chain alkylated sterol biosynthesis in the bacterial domain. However, this activity remains to be confirmed in a myxobacterial isolate.

The identification of three bacterial SMTs each capable of producing the 24-isopropyl side-chain via a single protein is also significant. Molecular clock analyses suggest demosponges were the likely source of Cryogenian 24-ipc biomarkers given that demosponges acquired a second SMT copy prior to their deposition, while pelagophyte algae did not acquire a third SMT copy until the Phanerozoic[36]. In addition, researchers have argued against rhizarian protists as a source of 24-isopropyl sterols given that *Bigelowiella natans* encodes only two SMT copies and is thus unlikely to synthesize 24-propyl sterols[20]. These interpretations rely on the hypothesis that multiple SMT copies are required to produce 24-isopropyl sterols. However, our results demonstrating one SMT is sufficient for side-chain propylation make SMT copy number an unreliable predictor of an organism's ability to produce 24-propyl sterols. Now that propylating SMTs have been identified, site-directed mutagenesis as performed with yeast[61,62] and plant[63] SMTs may reveal specific amino acid changes that allow an SMT protein to perform a third methylation. If such key residues are identified, they could act as a more reliable predictor of 24-isopropyl sterol biosynthesis directly from sequencing data than SMT copy number alone.

Finally, although we identified SMTs that produce 24-isopropyl sterols, proteins that methylate sterols to give other geologically important side-chain structures remain undiscovered. Identification of enzymes that methylate sterols at the C-26 position to give the 26-mes side-chain structure remains especially important when considering the biomarker evidence for Precambrian sponges. SMTs from sponges and bacteria produced (epi)codisterol suggesting both groups have the genomic capacity to produce substrates for hypothetical 26-SMTs[37,64]. However, it remains to be seen if sponges and bacteria, sponge-associated and otherwise, encode SMTs capable of methylation at C-26. If bacterial 26-SMTs are identified, verified side-chain alkylated sterol biosynthesis in a bacterial isolate would remain necessary in order to determine if extant bacteria produce 24-ipc and 26-mes sterol precursors in ratios corresponding to the $C_{30}$ sterane distributions observed in the Cryogenian. The identification of any unique isotopic signatures resulting from the reactions of 24- and 26-SMTs could also confirm bacteria as a contributing source of these and other side-chain alkylated biomarkers. Until then, eukaryotes remain the most likely source of side-chain alkylated steranes, and the co-occurrence of 24-ipc and 26-mes in high relative abundance continues to provide compelling evidence for derived sponges in the Cryogenian, consistent with molecular clock analyses[65–67]. However, the potential for sterol side-chain methylation by symbiotic and free-living bacteria should no longer be ignored both when analyzing the sterols of extant eukaryotes and when interpreting the presence of ergostane, stigmastane, and 24-isopropylcholestane in the geologic record.

## Methods
### Bioinformatics analysis
SMT homologs were identified in the Joint Genome Institute Integrated Microbial Genomes & Microbiomes (JGI IMG; https://img.jgi.

doe.gov), GenBank (https://www.ncbi.nlm.nih.gov/genbank), and Compagen (previously at http://compagen.org, now at http://compagen.unit.oist.jp) databases using the *Saccharomyces cerevisiae* S299C SMT amino acid sequence as the BLASTP search query (locus tag: YML008C, maximum e-value: $1e^{-50}$, minimum percent identity: 30%). The BLAST search of the JGI IMG metagenomic database resulted in 7966 SMT homologs as of May 2022. A subset of SMT homologs that contained the conserved methyltransferase and C-terminal domains and covered multiple environments were chosen for further analysis, with particular focus on SMT homologs from sponge microbiomes and those present in sterol biosynthesis gene clusters. Additional demosponge SMT sequences from *C. nucula*, *S. fasciculatas*, and *P. ficiformis* transcriptomes were provided by Gold[36]. Identifying sequence information is available in Supplementary Table 4. Conserved sterol methyltransferase C-terminal and methyltransferase 11 domains were identified using HmmerWeb.

Oxidosqualene cyclase homologs were identified through BLASTP searches of the metagenome scaffolds and MAGs containing the SMTs analyzed in this study and all bacterial and eukaryotic genomes available in JGI IMG as of October 2019 using the *Methylococcus capsulatus* Bath OSC protein sequence (locus tag: MCA2873, maximum e-value: $1e^{-50}$, minimum percent identity: 20%, minimum sequence length: 350 aa). Redundancy was decreased through manual removal of OSC homologs with 100% similarity at the amino acid level. The N-termini of fused SMO/OSC homologs were trimmed out of the dataset to begin the sequences at the squalene-hopene cyclase N-terminal domain as identified using HmmerWeb.

All protein sequence datasets were aligned via MUSCLE. Conserved SMT domains were aligned separately, then concatenated using Geneious. Maximum-likelihood trees were generated using IQ-Tree on XSEDE with 1000 ultra-fast bootstrap replicates using the best amino acid substitution models, gamma shape parameters, and invariable sites proportions under the Bayesian Information Criterion and Akaike Information Criterion using ModelTest-NG on XSEDE (Supplementary Table 5). Resulting phylogenetic trees were visualized in iTOL[68].

All contigs containing metagenomic SMTs were retrieved from JGI IMG or GenBank and analyzed with Whokaryote[69] using the default inputs with the minimum contig size decreased to 2000 base pairs. Calculated features and predictions are shown in Supplementary Table 2.

### Gene synthesis and molecular cloning
SMT DNA sequences were codon-optimized for expression in *E. coli* and artificially synthesized by GeneArt (ThermoFisher Scientific) or through the Department of Energy Joint Genome Institute (DOE JGI) DNA Synthesis Science Program. SMT sequences from DOE JGI were obtained in the IPTG-inducible plasmid pSRKGm-*lac*UV5-rbs5[70] in *E. coli* TOP10. SMT sequences from GeneArt were subcloned into pSRKGm-*lac*UV5-rbs5 by sequence and ligase independent cloning (SLIC)[71] and transformed into electrocompetent *E. coli* DH10B (Invitrogen) using a MicroPulser Electroporator (BioRad). Oligonucleotides were purchased from Integrated DNA Technologies (Coralville, IA). PCR was performed according the to the manufacturer's protocol using Phusion DNA Polymerase (New England Biolabs). DNA fragments were purified using the GeneJET Gel Extraction Kit (ThermoFisher Scientific). Plasmid DNA was isolated using the GeneJET Plasmid Miniprep Kit (ThermoFisher Scientific). Plasmid DNA was sequenced to confirm promoter and SMT sequences by ELIM Biopharm (Hayward, CA) using the following primers: 5'-AATGCAGCTGGCACGACAGG-3' (forward) and 5'-CCAGGGTTTTCCCAGTCAC-3' (reverse).

### Bacterial culture and heterologous expression
*E. coli* expression strains were cultured in 50 mL terrific broth (TB) supplemented with gentamycin (15 μg/mL) at 37 °C while shaking at 225 rpm. Cultures were induced with 500 μM isopropyl *β*-D-1-

thiogalactopyranoside (IPTG) at an $OD_{600}$ of ~0.6, then incubated an additional 4 h at 30 °C while shaking at 225 rpm. Cells were harvested by centrifugation at $4500 \times g$ for 10 min at 4 °C. Cell pellets were stored at −80 °C until sonication.

## Sterol methyltransferase assay

Cells pellets were resuspended in 5 mL buffer containing the following: 50 mM Tris-HCl, 2 mM $MgCl_2$, 20% glycerol (v/v), and 0.1% β-mercaptoethanol (v/v), pH 7.5. Cells were then lysed on ice via a Qsonica Sonicator Q500 equipped with a 3.2 mm probe at 30% amplitude pulsing at 5 s on, 15 s off for 8 min of total on time. Lysates were then partially clarified by centrifugation at $4500 \times g$ for 20 min at 4 °C. Total protein concentration in the resulting supernatant was quantified using a Coomassie (Bradford) Protein Assay Kit (Thermo Scientific) according to the manufacturer's Standard Microplate Protocol and a BioTek Synergy HT Microplate Reader. SMT assays were then immediately prepared using enough supernatant to give 4500 µg total protein. Other reaction components included 100 µM desmosterol (Sigma-Aldrich), 24-methylenecholesterol (Avanti Polar Lipids, Inc.), [28-$^{13}$C] 24-methylenecholesterol (Avanti Polar Lipids, Inc.), or synthesized [29-$^{13}$C] fucosterol or isofucosterol, 100 µM S-(5′-sdenosyl)-L-methionine chloride dihydrochloride (Sigma-Aldrich), and 0.1% Tween-80 (v/v) to a total reaction volume of 400 µL. Reactions were held at 30 °C for 20 h and then stored at −20 °C until lipid extraction.

## Lipid extraction

Lipids were extracted using a modified Bligh-Dyer method[72,73] with the completed in vitro reactions as the water phase. Reactions were sonicated in 10:5:4 (vol:vol:vol) methanol:dichloromethane:water for 1 h. The organic phase was then separated with twice the volume of 1:1 (vol:vol) dichloromethane:water followed by storage at −20 °C for >1 h. Following centrifugation at $2800 \times g$ for 10 min at 4 °C, the organic phase was transferred and evaporated under $N_2$ to give total lipid extracts (TLEs). TLEs were derivatized to trimethylsilyl ethers in 1:1 (vol:vol) pyridine:Bis(trimethylsilyl)trifluoroacetamide for 1 h at 70 °C prior to GC-MS analysis.

## GC-MS analysis

Lipids were separated with an Agilent 7890B Series GC equipped with two Agilent DB-17HT columns (30 m × 0.25 mm i.d. × 0.15 µm film thickness) in tandem with helium as the carrier gas at a constant flow of 1.1 ml/min and programmed as follows: 100 °C for 2 min, then 12 °C/min to 250 °C and held for 10 min, then 10 °C/min to 330 °C and held for 17.5 min. 2 µL of each sample was injected in splitless mode at 250 °C. The GC was coupled to an Agilent 5977 A Series MSD with the ion source at 230 °C and operated at 70 eV in EI mode scanning from 50 to 850 Da in 0.5 s. Data were analzyed using Agilent MassHunter Qualitative Analysis. Sterols were identified based on retention time and comparison to previously published spectra and laboratory standards. In vitro reactions for each SMT with both sterol substrates were performed at least twice, and the same sterol products were observed by GC-MS each time.

## NMR analysis

In addition to GC-MS analysis, structures of sterol products formed by the metagenomic sterol methyltransferases referred to as "Aplysina aerophoba JGICombinedJ30088_10000835 1 of 2" and "Chlamydiae sp., drinking water 1 of 2" were confirmed by NMR. $^{13}$C-labeling experiments were used to assist the identification of the biosynthetic products by 2D-NMR (heteronuclear single quantum coherence (HSQC) and heteronuclear multiple bond correlation (HMBC)) by increasing the amount of $^{13}$C at a specific position by two orders of magnitude compared to natural abundance. The two terminal isopropyl groups of 24-isopropyl sterols come from biosynthetically disparate origins, the isoprenoid pathway and SAM methylation. Because of the possibilities

of hydride-shifts and different positions of deprotonation, each of the four terminal methyl groups can have multiple different origins within the product[27]. In addition to enhanced sensitivity, $^{13}$C-labeling provided mechanistic details of a pathway which, because of the intrinsic symmetry of 24-isopropyl sterols, would be otherwise hidden.

Total lipid extracts from pooled in vitro reactions using [28-$^{13}$C] 24-methylenecholesterol or [29-$^{13}$C] isofucosterol as the substrate for both SMTs were fractionated using small silica pipet columns (1 mL) eluting with 5 mL each of hexanes:ethyl acetate 9:1, 2:1, and 1:1 (vol:vol). Steroidal constituents were found in the 2:1 solvent fraction and concentrated with a stream of $N_2$ prior to further purification by preparative thin-layer chromatography (TLC; 4:1 hexanes:ethyl acetate). Sterols were separated by high-performance liquid chromatography (HPLC)[39] and the fractions were analyzed by NMR[39]. Sterols were identified by comparison of the HSQC-DEPT and HMBC spectra with reference compounds.

In experiments with [28-$^{13}$C] 24-methylenecholesterol, the presence of isofucosterol was confirmed by a $^{1}$H-$^{13}$C cross peak corresponding to C-28 at 5.108 and 116.48 ppm. The presence of fucosterol was shown by an HSQC cross peak at 5.183 and 115.58 ppm. Ratios of isofucosterol to fucosterol were determined by comparison of the integral ratios of the cross peaks[39] for these two isomers. The presence of epiclerosterol was confirmed by cross peaks for C-28 at 1.290/1.380 and 26.01 ppm. Structures of the $C_{29}$ sterol products were confirmed by long-range HMBC correlations from the 29-methyl group. Isofucosterol showed an HMBC cross peak for H-29 and C-28 at 1.59 and 115.58 ppm; fucosterol and epiclerosterol showed HMBC cross peaks at 1.57 and 116.48 ppm and 0.799 and 26.01 ppm, respectively. The presence of 24S−24-isopropylcholesta-5,25-dienol labeled at the C-25 position was confirmed by HMBC from the 27-methyl group (cross peak between 1.570 and 147.38 ppm). The presence of 24-isopropylcholesta-5,24-dienol was confirmed by HMBC showing two adjacent cross peaks correlating the C-26 and C-27 methyl protons (1.633/1.651 ppm) with C-25 (123.1 ppm).

In experiments with [29-$^{13}$C] isofucosterol, the label distribution between the C-27 and C-26 positions of 24S−24-isopropylcholesta-5,25-dienol was shown by integration of the HSQC signals for C-27 (1.570 and 18.95 ppm) and C-30 (4.602/4.740 and 111.85). The label on the Z-vinylic methyl of 24-isopropylcholesta-5,24-dienol was detected by HSQC (1.651 and 19.73 ppm). 24-Isopropylcholesta-5,23-dienol was detected by label at 20.96 ppm exclusively, which is one of the two methyls of the Z-isopropyl group. The ratio of 24S−24-isopropylcholesta-5,25-dienol, 24-isopropylcholesta-5,24-dienol, and 24-isopropylcholesta-5,23-dienol was estimated by integration of the HSQC.

## Preparation of [29-$^{13}$C] fucosterol and isofucosterol

Specifically labeled $C_{29}$ sterol precursors were prepared by modification of the procedure reported in ref. 74. The precursor 24-formylcholesterol i-methyl ether was obtained either through hydroboration of i-Me 24-methylenecholesterol followed by Dess-Martin periodinane oxidation or ozonolysis of i-Me 24-vinylcholesterol. The latter reaction was carried out in 90:9:1 (vol:vol:vol) dichloromethane/methanol/pyridine and yielded the desired 28-formylcholesterol i-methyl ether and its methyl hemiacetal in a 1:5 ratio. The hemiacetal was isolated as a mixture of four isomers. $^{1}$H-NMR (600 MHz): 8.06-8.03 (1H, 28-OH), 4.68-4.63 (1H, 28-H), 3.56-3.52 (3H, 28-Ome) 3.332 (s, 3H, 6-Ome), 2.767 (s, 1H, 6-H), 1.108 (s, 3H, 19-H), 0.710 (s, 3H, 18-H). This mixture was reacted with $^{13}$C-methylmagnesium iodide to give four isomers of [29-$^{13}$C] i-Me 28-hydroxysitosterol: 24R,28R (H-28: 3.938 ppm; C-29: 21.97 ppm), 24S,28S (H-28: 3.938 ppm; C-29: 22.20 ppm), 24R,28 S (H-28: 3.820 ppm; C-29: 21.05 ppm), and 24S,28R (H-28: 3.803 ppm; C-29: 21.01 ppm). Preparative TLC (4:1 hexanes:ethyl acetate) gave two bands corresponding to a mixture of the RR and SS isomers (higher Rf) and a mixture of the RS and SR isomers (lower Rf) in ca. 2:1 ratio. These two mixtures were separately dehydrated with phosphorus oxychloride to obtain after deprotection [29-$^{13}$C]

fucosterol (C-29: 13.16 ppm) and [29-$^{13}$C] isofucosterol (C-29: 12.76 ppm), respectively.

## Reporting summary

Further information on research design is available in the Nature Portfolio Reporting Summary linked to this article.

## Data availability

The data that support the findings of this study are available within this paper and its supplementary information. GenBank accession codes and JGI IMG gene IDs for analyzed metagenomic SMTs are provided in Supplementary Table 4. JGI IMG gene IDs for the additional sterol biosynthesis homologs shown in Fig. 2 are provided in Supplementary Data 1. The raw extracted ion chromatogram data used to generate Fig. 1 are provided in Supplementary Data 2. The raw mass spectrometry data used to generate Supplementary Fig. 3 are provided in Supplementary Data 3. Raw NMR spectra are provided in Supplementary Figs. 4–7.

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

## Acknowledgements

We thank Prof. David A. Gold for providing demosponge SMT protein sequences, Jeremy H. Wei for technical assistance, and Hanon McShea for helpful discussions. The work (proposal: 503267) conducted by the U.S. Department of Energy Joint Genome Institute (https://ror.org/04xm1d337), a DOE Office of Science User Facility, is supported by the Office of Science of the U.S. Department of Energy operated under Contract No. DE-AC02-05CH11231. Funding for this study was provided by National Science Foundation Grant EAR-1752564 (to P.V.W.) and National Institute of General Medical Sciences Grant R15GM143714 (to J.-L.G.).

## Author contributions

M.O.B. wrote the manuscript with input from all co-authors. M.O.B. performed the molecular biology experiments, lipid extractions, and

GC-MS analyses. M.O.B. and P.V.W. performed the bioinformatic analyses. B.O.O. and J.-L.G. synthesized the [13]C labeled sterols and performed the NMR analyses. M.O.B. analyzed and interpreted the bioinformatic and GC-MS data. B.O.O. and J.-L.G. analyzed and interpreted the NMR data. P.V.W. and M.O.B. conceived the project with contribution from J.-L.G.

## Competing interests

The authors declare no competing interests.
