## [Peer Review File · Nature Communications]

Sterol methyltransferases in uncultured bacteria complicate eukaryotic biomarker interpretationsREVIEWER COMMENTS

Reviewer #1 (Remarks to the Author):

See pdf file attached for my review. I recommend publication after minor/major review.

Reviewer #2 (Remarks to the Author):

The earliest evidence for the evolution of animals is based on the occurrence of a specific sterane called 24-isopropylcholestane (24-ipc) in Precambrian rocks, which is supposedly derived from sponges. Several recent studies have either challenged or supported this claim, which in addition to sponges suggested algae as producers of 24-ipc. In their manuscript, Brown et al. describe a third option, bacteria living in symbiosis with sponges (as well as free-living bacteria), using novel methyltransferases to produce 24-ipc. The authors validate this hypothesis using a range of genetic and biochemical techniques, showing that these bacteria have functional sterol methyltransferases, even though the organisms are not available in culture. The authors thus conclusively demonstrate that bacteria can be sources of 24-ipc in addition to (or in symbiosis with) algae and bacteria. This study is of high quality and great value to a wide range of audiences including biochemists and geobiologists. I do not have any major criticisms. I have only one comment regarding the text: Some paragraphs are hard to read because the discussed taxonomy covers two domains and many phyla and other taxonomic levels. It is sometimes not always clear whether a specific taxon is a sponge, bacteria, or algae, so it would make sense to simplify or clarify the nomenclature wherever possible, such as by writing "pelagophyte algae" instead of "pelagophytes). Some additional suggestions are included below. The same arguably applies to the trivial names of sterols, which could either be fully explained when first mentioned or otherwise described.

Table 1: Can the presence/absence of other sterol biosynthesis genes such as squalene monooxygenase, oxidosqualene cyclase, the SMTs and others be included in the table to further support the presence of sterol biosynthesis pathways in these organisms?

Figure 1: Switch order of compounds V and IV in panel e?

Reviewer #3 (Remarks to the Author):

Dear authors,

Sterol methyltransferases (SMTs) had been previously identified in sponge genomes. Brown et al. for the first time identify SMTs in bacteria as well (sponge-associated bacteria and free-living freshwater bacteria). They expressed these sponge and bacteria enzymes in *E. coli*, and tested their activity showing that these SMTs are functional: not only can they add a methyl group on carbon 24 of the side-chain but the same enzyme can sequentially build an isopropyl side-chain on carbon 24, thereby helping to produce C30 sterol precursors of C30 biomarker 24-ipc. The significance of these results is high in the field of paleo geochemistry and evolution, where the origin of co-occurring biomarkers 24-ipc and 26-mes in the Neoproterozoic is highly debated. These biomarkers may represent the first signs of sponge presence (i.e. the first animals) where sterol precursors are well known, or may be produced by other organisms. Brown et al. show for the first time that bacteria have the capacity to produce 24-ipc precursors, and that only one SMT is necessary for the isopropyl group. This study will also definitely significantly move the field of sterol biosynthesis forward, whether it be in sponges or in bacteria.

The main weakness of this study is that Brown et al. have not explored sponge species

that actually produce high amounts of 24-ipc (or 26-mes) sterol precursors. All of the 8 sponge genomes investigated come from species which are not known to contain C30 sterols, so I was not surprised (as were the authors) their SMTs could not make C30 sterols. As for the metagenomes they come from sponge species with no C30 sterol precursors of 24-ipc/26-mes either. *A. aerophoba* does contain trace amounts of 24-ipc, but as many demosponges do (Zumberge et al., 2018. Table S5) and Brown et al. are aware of that (line 76). Altogether, sponge species from this manuscript are poor models to study the production of sponge biomarker precursors and eventually test the sponge biomarker hypothesis. Although it is clear the authors are aware of this weakness of the paper, it is only mentioned in the discussion (lines 163-165) although I think it should appear earlier in the main text. The authors need to explain from the start how the species sampled will limit their interpretations and conclusions on the "sponge biomarkers" per se. Also, the authors could add on Table 1 the known natural sterols of the species/genera investigated.

Since *A. aerophoba* only has trace amounts of 24-ipc precursors, the bacteria with the sterol gene cluster in *A. aerophoba* discovered by Brown et al. may therefore produce very little amounts of 24-ipc precursors, in disagreement with the amounts observed in Neoproterozoic rocks. So, yes, some bacteria may contribute to 24-isopropyl sterol biosynthesis in demosponges, or in the environment, but as for the Rhizaria (Nettersheim et al., 2019; Love et al., 2020) this contribution does not seem to be significant. If Brown et al. agree with this reasoning, I think it should be made clearer in their discussion, and in the title. Maybe the title of the ms could be tuned down; 'confounds' is a rather strong word in my mind, suggesting that the authors will present overturn some of the eukaryotic biomarkers interpretations. But the selection of species prevents a real challenge of the sponge biomarker hypothesis.

Here are a few other more specific remarks:

Lines 57-59. This sentence suggests that the species selected were good species to get insights into sterol 24-ipc precursors, but it is not that simple, as stated previously. The authors need to acknowledge at this point already the limits of using these sponge species.

Line 61-65. If I understood correctly, the sponge SMT was only shown to produce one alkylation at a time. How come? Since, SMT is producing 24-methylenecholesterol from desmosterol, why can't it continue to produce 24-ethyl sterols in one go (as the bacterial SMTs do)? I am missing in Fig 1c a chromatogram of an assay with a sponge SMT. Could the authors make this clearer in the manuscript?

Line 76. The authors should probably cite Zumberge et al., 2018, Table S5, instead of Zumberge's PhD thesis, or both.

Line 88. It is unclear how the 14 bacterial SMT were selected for heterologous expression and SMT assay. Can the authors explain this in the text? How many SMT were identified in the first place, are they that common? Did you notice if these SMTs were always found in sterol bacterial gene clusters? Since these are the first bacterial SMTs to be discovered I would give a bit more details about these.

Line 96 and 111, the authors mean '14' not '24', right?

Line 159. Here the authors must specify that the sterols of *Oscarella carmela* and *Sycon ciliatum* have not been analysed so the "previous hypotheses" mentioned are based on the possession of a single SMT copy (Gold et al., 2016). Maybe these two sponge species only naturally produce 24-methyl sterols, as the assay suggests. But yes, I agree that depending on the species, sponge SMTs seem to add one or several methyl groups on carbon 24.

Line 171-172. Although a valid hypothesis, I do not see how your data can suggest that sponges and bacteria "may therefore be required to produce 24-isopropyl sterols"

together. Especially since you have not studied a STM from a sponge with high propylated sterols.

Line 200-205. Suggested improvement: have the authors tried to detect Chlamydiae in the metagenome of *Aplysina aerophoba*?

I would like to point this recent preprint claiming to have found substantial diversity of Chlamydiae in sponges (*Halichondria* and *Haliclona* species). They also identified sterol reductases that they suggest could have been acquired by HGT from other bacteria or from eukaryotes.

<https://www.biorxiv.org/content/10.1101/2021.12.21.473556v1.abstract>

Line 356. Give web link to the JGI IMG.

Line 357. I could not access the website www.compagen.org. Maybe I was unlucky, but please double-check that the link is working.

Line 403. It is not clear in the text if replicates of these assays were done? at least for some of the SMTs.

Reviewer 1:

This manuscript revolves around an interesting and well executed set of experiments and investigation, encompassing bioinformatics, gene synthesis and lipid analyses using NMR and GC-MS. The study helps inform us at how bacteria (mainly symbionts/parasites) and sponges may modify the sidechain of sterols to produce 24-alkylated steroids. The focus of the take-home message seems to be that bacteria complicate and/or contradict conventional wisdom about eukaryotes being the major ancient sources of 24-alkylated steroids. There was also some intriguing evidence obtained that bacterial symbionts could play a role in synthesis of unconventional sterols in sponges.

We thank the reviewer for their thoughtful and constructive comments. We have attempted to address many of the concerns raised and hope they will find the revised manuscript acceptable.

Some of the structural transformations of sterol model compounds that were observed with isolated SMTs from bacterial symbionts of sponges and with a bacterial parasite (*Chlamydiae* sp.), is of importance and of great interest to biomarker geochemists. There are lots of intriguing results reported in this paper. But, there is still no good evidence that free-living bacteria can make these 24-alkylated regular sterols in high abundance as their major membrane lipids. Unlike for demosponges, other eukaryotes, etc. Yes, it is early days to expect us to know the environmental impact or cellular abundance of bacterial sterols but, at the same time, the authors should avoid over-extrapolating the implications of their findings.

I have some concerns that the take home messages (Abstract and Title) and implications overplayed for the general reader, As written, a non-specialist might cite this work that it shows clear evidence that bacteria are major biogenic source of 24-alkylated steranes, and of 24-ipc more specifically,. But, there is still no concrete evidence that bacteria can make 24-alkylated sterols de novo and in abundance as their major sterols.

We agree with the reviewer on the first point – we do not want readers to misinterpret our findings as unequivocal evidence that bacteria are a major biogenic source of 24-alkylated sterols. We have changed the title to “Sterol methyltransferases in uncultured bacteria complicate eukaryotic biomarker interpretations” and adjusted the abstract to reflect this.

On the second point, we agree that the evidence is not concrete that bacteria can make 24-alkylated sterols de novo as we do not have any bacteria with SMTs in culture. But we would argue that the sterol biosynthesis gene clusters containing functional SMTs we identify provide enough evidence to suggest the possibility. We do not hope to rewrite the entire sterane record with this work, but rather make it known that bacteria should no longer be ignored as potential sources of 24-alkylated steranes. We have adjusted the abstract and discussion to make this clearer.

Any good candidate source organism for 24-ipc must i) have appeared in the Neoproterozoic (between 660 and 635 Ma) consistent with first appearance of 24-ipc in rock record, and ii) make these as major sterols in their cells, not just as minor/trace constituents that would be diluted by other steroid compounds and not detected in the ancient sedimentary record. Demosponges fit the bill for both criteria (and multiple animal molecular clocks predict a Tonian or Cryogenian divergence for demosponges) while no bacterium is ever known to make 24-ipc as a major membrane sterol. That is the bottom line as it stands. There may well be a possible role for bacterial symbionts working in tandem with the sponge host to make abundant 24-ip, and other unconventional sterols, I agree with this though. .

Any good candidate source organism for ancient 24-ipc steranes must i) have appeared in the Neoproterozoic (between 660 and 635 Ma) consistent with first appearance of 24-ipc in rock record, and ii) make these as major sterols in their cells, not just as minor/trace constituents that go undetected in the ancient sedimentary record. Demosponges fit the bill for both criteria (since various animal molecular clocks predict Neoproterozoic divergence age for demosponges) while no known bacterium is known to make 24-ipc as a major lipid of its own volition.

We explicitly state in the discussion that 24-ipc (and 26-mes) still provide compelling evidence for demosponges in the Cryogenian. We have added text stating this is consistent with molecular clocks.

Indeed, the timing of the appearance of 24-alkylsteranes in the rocks record in the Neoproterozoic Era, strongly argues against a direct biogenic source from bacteria since bacterial terpenoids (such as hopanes and lanostanes) are found as far back as 1.64 billion years ago. Why do these regular steranes not appear in the geological record until 1 billion years after the oldest known bacterial markers are found in immature rocks? This evidence is much more consistent with a eukaryotic source of 24-alkylated steranes than with a bacterial source. In fact, the timing of first appearance argues persuasively against a direct bacterial origin.

I would like to see this work published, as the manuscript is based around an interesting set of experiments that have been well designed and executed. If the authors can be a bit less gung ho in extrapolating their findings at certain places in the manuscript (see Main Points) then I would recommend publication of this paper in Nature Communications pending minor revisions.

Again, we thank the reviewer for their extensive and constructive comments. Overall, we feel their input has improved our manuscript significantly.

Main points:

1) Abstract

In the Abstract, the text reads "...three bacterial methyltransferases each capable of sequential methylations resulting in the 24-isopropyl side-chain" come from 3 bacterial symbionts of sponges and a bacterial parasite. This obviously complicates the interpretation with respect to the sponge biomarker hypothesis as the symbionts and parasite live intracellularly within eukaryotic hosts and are not freeliving bacteria. So, in Abstract amend the text to: "and identify three bacterial methyltransferases from symbionts/parasites each capable of sequential methylations resulting in the 24-isopropyl side-chain"

We rephrased this statement to make it clear that the 3 propylating bacterial SMTs are from symbionts.

Another example of text that needs edited is the last sentence of the Abstract, since there is no known bacterial species that can make regular 24-alkylated sterols de novo as major membrane lipids and finding bacterial SMTs in metagenomes tells us nothing about the sterols being made or their environmental abundance. The last sentence also confusingly conflates the enzymatic capacity to make 24-alkylsteranes per se, with specific synthesis of 24-ipc and this whole sentence should be reworded.

We have rephrased the last two sentences in the abstract as follows for clarity:

We demonstrate that bacteria have the genomic capacity to synthesize side-chain alkylated sterols, and that bacterial symbionts may contribute to 24-isopropyl sterol biosynthesis in demosponges. Together, our results suggest bacteria should not be dismissed as potential contributing sources of side-chain alkylated sterane biomarkers in the rock record.

2) The main novel findings of the study are shown in Figure 1, sterol transformations from SMTs from sponges and bacteria:

a) Note, none of the bacteria here are free-living bacteria. The bacteria used are either sponge symbionts or a freshwater parasite (*Chlamydiae* sp.) that live intracellularly within eukaryotic hosts. As such, the results cannot be readily extrapolated to free-living bacteria. Plus, the use of bacterial symbionts of sponges complicates testing of the sponge biomarker hypothesis. The methylation conversions of 24-methylenecholesterol model compound into C29 and C30 sterols looks extremely low, gauged visually from the relative peak areas of the different sterol compounds from GCMS, even under the optimized laboratory conditions that increases sterol solubility and hence the sterol reactivity.

We have added chromatograms to Figure 1 for an SMT from a *Sandaracinus* sp. MAG (a free-living *mxobacterium*) that we had demonstrated is able to ethylate at C-24 to clarify that some of the functional SMTs are from free-living bacteria as we agree this

wasn't clear from Table 1 alone. We have also added text to the discussion addressing this.

b) There was one surprising finding for the *Chlamydia* fed with desmosterol (Figure 1c, middle trace) that one SMT can produce three sequential methylations such that an isopropyl group can be added. This seems like a major finding of the study. But bear in mind, *Chlamydia* doesn't make any sterols of its own volition. So, the sterol patterns found here with isolated SMTs and sterol model compounds don't necessarily represent the sterol distributions of the parent organism (see later).

We agree with the reviewer that we have no evidence that *Chlamydiae* produce sterols *de novo* and do not make that claim in the text.

c) Why was 24-methylenecholesterol not fed to *Chlamydiae* sp. to complete the symmetry of the experiments in the 6 panels in the figure?

It was, and we have added this chromatogram to Figure 1. Each SMT was tested with both desmosterol and 24-methylenecholesterol.

3) Line 66: Heading : amend to "Three Bacterial SMTs from symbionts/ parasites produce 24-isopropylsterols from sterol model compounds". A more accurate representation of the findings.

We changed the heading to "Three SMTs from symbiotic bacteria produce 24-isopropylsterols". We did not add "from sterol model compounds". Giner and Djerassi (1990) showed desmosterol and 24-methylenecholesterol are suitable substrates for side-chain methylation in sponges and in 1991 showed the pelagophyte pathway to 24-propylidenecholesterol proceeds through desmosterol and 24-methylenecholesterol. Further, a recent preprint (Michellod et al., 2022, *bioRxiv*) showed that the annelid SMT pathway proceeds from desmosterol to 24-methylenecholesterol to phytosterols. We have added additional text elaborating on these previous studies in the results section.

4) Figure 2 schematic and caption:

Note, *Petrosia ficiformis* and *Theonella swinhoei* do NOT make 24-ipc sterols. Indeed, *Theonella* doesn't seem to make regular (4-desmethyl) sterols amongst its major characteristic sterols.

The SMTs in these gene clusters did not produce 24-ipc sterols in our experiments so we do not suggest bacteria make 24-ipc sterols in these sponges. We have noted the major sterols of these species in Supplementary Table 1.

Aplysina aerophoba only makes 24-ipc as a minor sterol (it's main 2 sterols in the sponge biomass are 26-methylated compounds, which you didn't detect with the SMT isolated from the microbiome).

We have expanded on the major sterols of *Aplysina aerophoba* in the discussion.

Which downstream sterols are you matching in this figure to the main text ? It seems disconnected.

This is potentially confusing do you specifically mean 24-ipc sterols and/or 24-alkylated sterols, when in reality the schematic diagram shows neither of these two entities.

We mean 24-alkylated sterols broadly and have adjusted the title to reflect that. We also expanded the pathway to include side-chain methylation in the figure.

Your schematic shows only 4-methylsterols as the downstream sterols, but these are not regular (4-desmethyl) sterols. These are not the typical “eukaryotic sterols” as referred to in the title that is meant to be the subject matter here.

Please clarify what you mean with this Figure and caption as it seems somewhat disconnected to the main text.

We have added complete demethylation at C-4 to the pathway in Figure 2 and adjusted the caption, as well as included a clarification on complete demethylation at C-4 in bacteria in the discussion.

5) Figure 3 caption

The ML tree constructed contains no species that make 24-ipc sterols as major sterols. Just clarify in the caption that the SMTs shown cannot be assigned to the synthesis of any specific sterols or inform us about their environmental and/or cellular abundance. It is not predictable what distribution of sterols may be made from only finding SMT genes.

SMT sequences from species that make 24-ipc sterols as major sterols are not publicly available as explained elsewhere in the manuscript. We do not claim that the position of SMTs in the tree allows us to predict the synthesis/distribution of specific sterols or informs us about their environmental and/or cellular abundance. The purpose of the SMT tree, as described in the results, is to provide evidence that 1) the metagenomic SMTs we tested are bacterial and 2) SMTs from bacterial sponge symbionts were not acquired from the sponge hosts.

It needs clarification that finding an SMT does not imply 24-ipc synthesis. Indeed, it doesn't tell us about which sterol compounds are made, if any.

We have added symbols to the SMT tree denoting the number of carbons each SMT added in our experiments. We also made it clear in the caption that the SMTs in the tree were tested in vitro.

6) Failing to utilize the age-selective temporal patterns of Precambrian steranes as part of the evidence for evaluating the most parsimonious explanation is a weakness of the paper. How does a proposed bacterial origin of Precambrian steranes tie in with the timing of their first appearance in the rock record?

So, if SMTs originated in bacteria it begs the question of why don't we see 24-alkylated steranes in the geological record as far back as 1.64 Ga (Barney Creek Formation, Australia)? Regular steranes (initially dominated by cholestane) are not routinely detected till ca. 820 Ma and younger, as the record of eukaryotic microfossils begins to diversify. The first robust finding of ergostane (24-methylcholestane) is from low maturity sedimentary rocks from the Visingsö Group of Tonian age, estimated at ca. 780 Ma and younger (Zumberge et al., 2019, *Geobiology*). This is almost a 1 billion year-lag in time between the oldest well-preserved bacterial biomarkers (1.64 Ga) and the oldest occurrence of 24-alkylated steranes (0.78 Ga).

We do not claim that SMTs originated in bacteria (although others have - Hoshino et al., 2021), but we can see the potential for confusion based on the original title of Figure 3. We have changed the title to: Functional metagenomic SMTs are bacterial.

We also do not claim that bacteria are the source of the earliest 24-alkylated steranes to the exclusion of eukaryotes because we agree that the evidence as it stands does not support this. We merely hope to impart the idea that bacteria should no longer be excluded as potential contributing sources of 24-alkylated steranes in general as we have provided biochemical and genomic evidence of this possibility. We have adjusted text in the abstract and discussion to better reflect this.

7) Table 1 caption: you need to clarify that this is the sterols distribution obtained from desmosterol/whatever else in laboratory experiments using sterol model compounds and with isolated SMTs, not representative of sterol distributions from natural specimens or environmental samples.

The sterol distributions from these experiments do not necessarily reflect the main sterols found in the parent organisms (which leads to a lot of questions in itself, e.g., *Aplysina* sponges will have aplysterols as their main primary sterols in natural specimens).

We have added "in vitro" to the end of the caption to clarify this.

8) As far as as I am aware, there still is no evidence that any bacterial isolate, from any species, can make any C30 sterols as major lipids de novo? Does this still hold?

Yes, the reviewer is correct there is no evidence that any bacterial isolate can produce C30 sterols.

If the answer is “yes”, then this should be explicitly stated in the text and brought into the Discussion section for better balance and perspective when discussing a bacterial source for Neoproterozoic 24-ipc steranes.

We have expanded our discussion on the lack of SMTs in bacterial isolates throughout the text to make this clearer.

The result of finding bacterial SMTs in environmental metagenomes tell us nothing about the bacterial capacity to make 24-alkylated sterols de novo and/or which particular sterol compounds can be made, and neither does it inform us about their quantitative contribution to the environmental lipid record.

We respectfully disagree with the reviewer. We have identified bacterial SMTs in environmental metagenomes, provided biochemical evidence that they are functional, and provided genomic evidence of their association with other sterol biosynthesis genes. These multiple lines of evidence indicate that the bacterial capacity to make 24-alkylated sterols de novo is likely. We agree that we cannot know which specific sterol compounds are made or their quantitative contribution to the environmental lipid record until we have a bacterium in culture making 24-alkylated sterols. As such, we have avoided suggesting major reinterpretations of the Neoproterozoic 24-alkylated sterane record or making new interpretations ourselves.

9) Regarding how meaningful the sterol model compound/SMT actually are with respect to predicting sterol contents of parent organisms:

Whether these sterol model compound/SMT incubation experiments, with detergents in the media etc, are representative of the reactivity and biological availability naturally occurring marine sterols (found mainly sequestered in complex particulate and/or dissolved natural aquatic environments is open to question. They probably give insights into “best-case” scenarios even though they likely exaggerate the enzymatic and chemical reactivity of marine sterols. But, they can give misleading information about which sterols are expected to be found in the parent organisms. For example, *Chlamydiae* here that doesn't make its own sterols but has an SMT that can produce 24-ipc (if manipulated with model compounds in the lab).

Generally, in vitro enzyme assays are not best-case scenarios of enzymatic activity. The best-case scenario for any enzyme is in the cell in its native environment. Therefore, we agree with the reviewer that these enzymatic assays do not necessarily reflect the exact sterol patterns present in the parent organism. But that is not what we are saying in the manuscript. Our results demonstrate the reaction carried out by these proteins, the substrates they utilize, and the products they produce – in vitro assays such as these are well-established methods used to confirm the activity of a protein. To fully assess the lipids produced by the parent organism would require in vivo experiments including extensive lipid analyses. This is not possible as we do not yet have these organisms in culture. Given these limitations, genomic assessment coupled

to in vitro assays is currently our best evidence for the function of these proteins. That was the goal of these experiments – not to claim that we could assess the full lipid profile of the parent (uncultured) organism from these in vitro experiments alone.

Indeed, it could be argued that the laboratory conditions used for alkylation of sterol model compounds are not representative of the conditions in most aquatic natural environments as the main organic matter substrates would not be free sterols...it would be particulate and dissolved organic matter that is structurally complex and of high molecular weight. It is not clear if bacterial SMTs would work on such complex substrates? The enzymes might not recognize the sterol compounds in such an organic matrix.

It appears the reviewer might be suggesting that the results of in vitro assays performed in a test tube with cell lysates of a heterologous host engineered to express SMTs are somehow evidence for bacterial methylation of exogenous sterols in the environment. We do not think our in vitro assays suggest this and do not make these claims in our manuscript.

That would make a bacterial methylation of exogenous sterols much less likely in the natural environment than in these laboratory conditions. Do the authors agree or disagree with this notion?

We agree that the methylation of exogeneous sterols (i.e. those found outside cells in particulate and dissolved organic matter) by bacteria would be challenging so we did not propose this as a possibility.

10) The demosponge species that make 24-ipc as major cell membrane sterols (as their most abundant monohydroxy sterols) have been widely known in the scientific literature, over the course of 40 years. It is not difficult to hunt a slew of papers on this from a google search and include many closely related species from the genera *Cymbastela*, *Topsentia*, *Petromica*, *Halichondria*, *Ciocalypa*.

We were unable to find genomic data for these species. If it does exist, it is not publicly available. There was a *Cymbastela stipitata* transcriptome deposited into GenBank after our JGI synthesis grant that funded our synthesis of various SMTs was fulfilled but to our knowledge, sterols have not been reported from that species.

It puzzles me why molecular biologists seem to always seem to avoid looking at demosponge species from this list? Wouldn't one of these demosponge species be a priority? Rather than just stacking up evidence for complications with bacterial steroid synthesis?

We are not avoiding demosponge species from this list. Genomic data for these species do not exist or are inaccessible.

All the sponges (and sponge symbionts) used in this investigation are from species that do NOT make 24-ipc amongst their major sterols.

The reviewer is correct. We have tried to make this clearer in the manuscript and added this to Supplementary Table 1.

This needs rectified with a study using a more appropriate demosponge species (host and/or symbionts) as a model organism. This has been obvious since the Gold et al., (2016) study where *Petrosia ficiformis* was used and this is a demosponge that makes petrosterol (an unusual C29 sterol) but not 24-ipc sterols in any significant amounts.

We agree with the reviewer as stated in the discussion.

I am puzzled by this apparent reluctance to work on a more appropriate model demosponge species for studying 24-ipc synthesis. That's now 6 years of poor choices for demosponges to study 24-ipc sterol synthesis.

If the sequencing data were available to us, we would have included it in our analyses. We hope the results of our study encourage experts in holobiont sequencing to take on this work.

Importantly, I think a distinction should be made between organisms where 24-ipc compounds are synthesized as major eukaryotic membrane sterols (certain groups of demosponges) versus the capacity to make these compounds as trace by-products from methylation of environmental sterol model compounds under carefully selected laboratory conditions (with SAM added, detergent added to overcome the low aqueous solubility). The mechanisms (especially the role of animal host vs symbiont SMT in 24-ipc synthesis) could be significantly different for both scenarios.

We explicitly state in the introduction that demosponges are the only organisms known to produce 24-ipc as their major sterols. It is widely understood in molecular biology and biochemistry that the results of in vitro experiments do not necessarily reflect what occurs in vivo. Most often, enzymes do not work as well in vitro as they do in vivo, even with optimization. We explicitly state in the discussion that we cannot know if bacteria make high amounts of 24-ipc until we have an organism making 24-isopropyl sterols in culture.

11) Discussion, paragraph, lines 197-209

This whole paragraph needs revised since the authors are mixing up 24-ipc with 24-npc steranes.

They cite Ref 47 Moldovan et al., 1984.

It is stated that 24-ipc steranes have been used as an environmental indicator of marine origins or marine incursions. This isn't strictly correct since it is the 24-npc sterane, made mainly by marine pelagophyte algal ancestors since the Devonian Period, that has been widely used for this paleoenvironmental assessment. So, this is a completely different C30 compound to 24-ipc and the distinction should be recognized. A couple of papers by Wunsche et al. (1987) may say that they expect 24-ipc to be associated with marine conditions but 24-ipc has not been widely applied as a marine marker by organic geochemists..

A later paper by Moldown et al. (1990) Science confirms that it is specifically the 24-n-propylcholestanes (aka 24-npc) that have been widely used as a marine environmental marker NOT 24-ipc . This presence or absence of 24-npc generally works well for Devonian and younger rocks and oils .

So, because of this misunderstanding, this entire paragraph ends up being a bit misleading and it should either be deleted or undergo major revision.

Note, that the Wünsche et al. (1987) paper on sediment sterols from a small freshwater pond was also referenced and discussed in the SI document accompanying the Love et al. (2009) Nature paper. For convenience, I have cut and posted relevant text from this:

“A report of significant amounts of 24-isopropyl-5 α (H)-cholest-22-en-3 β -ol (although this compound structure was only “tentatively identified”) in a Recent freshwater lake sediment^{S81} “

The uncertainty in the structural assignments for the 24-ipc compounds in this 1987 study casts doubt on whether this was a genuine 24-ipc sterol signal though.

Indeed, C30 steranes or other C30 steroids are not commonly found in freshwater-sourced sediments/oils and coals, this generality holds for the biomarker across geological time (Fu Jiamo et al., 1990, OG; Horsfield et al., 1994, OG). C27-C29 steranes are commonly detected in lake sediments but it is 4-methylstigmastanes that are the most common C30 compounds in low salinity settings, but these are not regular C30 steranes.

We thank the reviewer for this clarification. We have removed the portion of the discussion on Chlamydiae and 24-ipc as a marine biomarker and instead focus on Chlamydiae, sterol modification, and sponges as suggested by Reviewer 3.

12) The authors should consider analysis of sterols in shallow sediments from different modern sedimentary environments, including freshwater lakes. I would wager that C30 regular steranes/sterols are not going to be commonly detected lipid constituents but that C29 compounds will be abundant. There is the one Wunsche et al., study on a puddle in 1987 that seems to indicate some 24-ipc sterol (though the 24-ipc compound assignments was only tentative).

If the bacterial methylation of phytosterols is a prominent and ubiquitous process, then 24-ipc sterols should show up as detectable analytes in many modern aquatic environments as sedimentary lipids. If they don't then there is a problem with the general hypothesis that C30 sterols are derived from diagenetic modification of C27-C29 sterols, which would involve bacterial methylation of exogenous sterols.

We do not suggest that bacterial methylation of phytosterols/exogenous sterols is prominent and ubiquitous because there are not data to support that it occurs.

It is an easy thing to test...are 24-ipc sterols detectable and abundant in range of modern aquatic environments from water column particulates and sediments? Thus far, I have never seen 24-ipc as an abundant sterol component of any modern lake or marine sediment. We have worked on microbial mat ecosystems, and sediments that are deposited beneath active benthic mat layers in Mexico that have grown in carefully maintained salt ponds for decades and 24-ipc and/or 26-mes was not a significant sterane constituent in any sedimentary layers as either a free or kerogen-bound constituents (Lee et al., 2019, 2021, OG).

This is a very interesting suggestion, but we feel testing if 24-ipc sterols are detectable and abundant in range of modern aquatic environments from water column particulates and sediments is beyond the scope of this study.

13) So, let's compare the three most likely source input/mechanistic possibilities which have been proposed to explain the appearance and abundance of 24-ipc in some, but not all (Ediacaran rocks from Baltica do not contain 24-ipc, Pehr et al., 2018, Nat. Comms), rocks and oils of Cryogenian-Ediacaran age.

i) Direct biogenic origin from demosponge

* Multiple animal molecular clock studies consistently predict that demosponges diverged in Neoproterozoic (see refs in Zumberge et al., 2018, Nature EE), so the timing of their appearance is consistent with first appearance of 24-ipc between 660-635 Ma as a new biogenic source of steroid lipids. Please mention the molecular clock constraints (not just the Gold et al., 2016, SMT gene tree) for demosponge divergence more generally somewhere in the text.

We have added text to the discussion mentioning molecular clock constraints.

* Multiple demosponge species (many closely related species from the genera Cymbastela, Topsentia, Petromica, Halichondria, Ciocalypta) produce 24-ipc as their major cell membrane sterols in high absolute abundance, so the source organismal potential for being the source of sedimentary 24-ipc is high for demosponges.

We explicitly state that demosponges are the only organisms known to contain 24-isopropyl sterols in the introduction.

These are not just trace sterol constituents and 24-ipc can be found in different sponge cell types as major lipids located in the sponge cell membranes as sometimes the dominant sterols (Lawson et al., 1988, Lipids; Zimmerman et al., 1989, Lipids).

ii) Direct biogenic origin from bacteria

* No evidence that any bacteria can make 24-ipc de novo as major membrane lipids

Our work provides biochemical and genomic evidence for de novo biosynthesis, but there is no evidence that bacteria can make 24-ipc as major membrane lipids as there are no bacteria with SMTs in culture.

*Bacterial terpenoid biomarkers (hopanes, from squalene cyclization) are found in rocks as old as 1.64 billion years old from Barney Creek Formation....why didn't 24-ipc appear earlier in the Precambrian record than 660-635 Ma if bacteria were the most probable source?

While SHC and OSC are homologous enzymes, hopanoids are not sterols and the downstream biosynthetic pathways are quite different. We would therefore argue that the presence of hopanes at 1.64 Gya does not negate bacteria as potential contributors to the Neoproterozoic 24-alkylated sterane record.

On the lanostanes mentioned earlier in the review: SMTs in extant organisms generally work poorly on lanosterol and typically utilize substrates fully demethylated at C-4 and C-14 (if not cycloartenol or 24-methylenelophenol as in plants). It would therefore be a reasonable hypothesis that bacterial SMTs did not originate or become widespread until after the evolution of lanosterol demethylation genes, which could have been millions of years after the origin of OSC in bacteria. This would be consistent with the Bloch hypothesis that sterol biosynthesis pathways were gradually optimized over time.

Further, this argument could also apply to eukaryotic sterol synthesis. Gold's 2016 molecular clock predicts the ancestral SMT originated at ~1700 Ma. So why do we not observe 24-alkylated steranes before ~660 Ma? If the first cholestanes, attributed to eukaryotes, appear at ~800 Ma, why are 24-alkylated steranes absent until ~660 Ma?

*This discrepancy with the temporal biomarker record makes a bacterial source highly unlikely.

We appreciate the reviewer's interpretation that this discrepancy makes a bacterial source unlikely, but respectfully disagree. It takes time to evolve new

enzymes, and more time for those enzymes to become widespread enough to leave their trace in the geologic record.

iii) Alteration of algal sterols through side-chain methylation

* Could be made feasibly from microbial methylation of C₂₉ algal sterols, and the Neoproterozoic timing works for appearance of both C₂₉ and C₃₀ steranes.

We are unaware of any evidence for microbial side-chain methylation of exogenous C₂₉ algal sterols.

* Seems a better bet as an alternative to sponges compared with de novo synthesis from bacteria

We respectfully disagree.

* Still basically represents a eukaryotic source for these sterane compounds (from exogenous sterols).

If there was evidence of microbial side-chain methylation of exogenous C₂₉ algal sterols, yes.

* This mechanism falls down since predicted C₃₀ sterane compounds that should be produced from this mechanism (for example, 22methystigmastane and/or 23-methystigmatane from methylation of stigmasterol) are not found. It is 24-npc, 24-ipc and 26-mes steranes (3 compounds maximum out of many structural possibilities) that are found in but in widely different relative proportions from sample to sample. All three sterane compounds can be made by sponges,

We are unaware of any evidence of microbial methylation at C-22 or C-23.

* A second major problem is some Ediacaran rocks with C₂₇-C₂₉ steranes, and a large C₂₉ sterane predominance, actually yield no detectable C₃₀ steranes (Ediacaran rocks from Baltica used in Pehr et al., 2018, Nat. Comms). So, there is not a common background signal of C₃₀ steranes in terms of common distributions and abundance. This also holds for Ordovician rocks (Rohrsen et al., 2015, OG), which generally contain high C₂₉ sterane signal but trace/nil of C₃₀ steranes.

We are unclear as to the point of this comment. Is this not referring to microbial methylation of algal sterols but rather the abiotic methylation of algal sterols during diagenesis? If so, we mentioned that in the introduction. Further, we explicitly state in the discussion that 24-ipc is still compelling evidence for demosponges.

iv) Rhizaria (Nettersheim et al., 2019, Nature EE) are a poor candidate group as source organisms and hardly mentioned these days by the Brocks and Hallmann groups who first proposed this. So, I wouldn't list this as a viable option.

*They reported only traces of 24-ipc and/or 26-mes (typically <0.01% of total steranes) made as artifacts by their PtO₂ hydrogenation method from methylation of C₂₉ sterols as trace byproducts. So, these are unlikely to be derived from primary sterols. Way below the abundances that are found in the Neoproterozoic record (1-4% of total C₂₇-C₃₀ steranes and getting as high as 13% in South Oman rocks).

*No intact 24-ipc or 26-mes sterols have been reported, indeed Leblond et al. (2005) found mainly C₂₈ and C₂₉ sterols but no traces of C₃₀ sterols in chloroachniophytes (photosynthetic rhizarians). Whether these eukaryotes use 2 SMTs to make C₂₉ sterols is not confirmed, but they don't seem to make appreciable primary C₃₀ sterols in any case.

We agree that Rhizaria are a poor candidate group as source organisms and understand that the authors of the Nettersheim 2019 work no longer mention them. However, others in the field still bring them up (see the 2022 preprint on annelid SMTs from Gold and colleagues), and we feel it's necessary to summarize all the recently published work on C₃₀ sterols. We therefore chose not to remove the mention of Rhizaria, but we have added additional text to the introduction to make it clearer that they are not a good potential source.

14) What do you want the take-home message of this paper to be to the general reader? That bacteria are a more likely source of Neoproterozoic 24-alkylsteranes and 24-ipc than eukaryotes? That is a very hard sell given all the available evidence on the table.

We agree with the reviewer and do not state in the manuscript that bacteria are a more likely source of Neoproterozoic 24-alkylsteranes than eukaryotes. We show for the first time that bacterial SMTs can alkylate sterols and that a single SMT from bacterial sources is capable of propylating at C-24. This is unexpected and novel biochemistry – and bacterial enzymes, not eukaryotic ones, are the only ones capable of this. This evidence is significant as it suggests that bacteria are capable of producing 24-alkylated sterols and should not be ignored as contributing sources. This is what we want the take-home message of the paper to be to the general reader. We have adjusted the title, abstract, and discussion to make this clearer.

There is no evidence that any free bacteria can produce 24-ipc sterols as major lipids i.e. making these as membrane lipids and as the major sterols.

We have expanded the discussion to make this clearer.

Clearly, the points discussed in 11) above, then the temporal patterns still favor a eukaryotic origin by far. That's for 24-alkylsteranes per se, and not just 24-ipc, in my opinion.

We appreciate the reviewer's opinion.

15) I want the authors to be a bit more explicit regarding what they think the most likely important role of bacterial involvement is with respect to sources of Precambrian 24-ipc and 24-alkyl sterane”:

a) bacterial methylation of acquired environmental sterols (side-chain alkylation), versus b) de novo bacterial synthesis , versus c) bacterial symbionts within eukaryotic hosts?

Please clarify.

a) There is no evidence of bacterial methylation of acquired environmental sterols, and this would be biochemically challenging as argued by the reviewer early. We therefore do not present this as a possibility. b) and c) are both plausible based on the results of this work and of previous studies, so we discuss both possibilities. We have elaborated on both possibilities in the discussion for clarity.

It seems to me that any suggestions of bacterial de novo synthesis are currently unfounded.

We respectfully disagree with the reviewer. Biochemical evidence of functional bacterial SMTs and genomic evidence of other sterol biosynthesis genes associated with these SMTs, as we have provided, suggests de novo synthesis is likely. We do not have an organism in culture producing 24-alkylated sterols so we cannot say this with certainty. We have tried to make this clearer in the discussion.

Reviewer 2:

The earliest evidence for the evolution of animals is based on the occurrence of a specific sterane called 24-isopropylcholestane (24-ipc) in Precambrian rocks, which is supposedly derived from sponges. Several recent studies have either challenged or supported this claim, which in addition to sponges suggested algae as producers of 24-ipc. In their manuscript, Brown et al. describe a third option, bacteria living in symbiosis with sponges (as well as free-living bacteria), using novel methyltransferases to produce 24-ipc. The authors validate this hypothesis using a range of genetic and biochemical techniques, showing that these bacteria have functional sterol methyltransferases, even though the organisms are not available in culture. The authors thus conclusively demonstrate that bacteria can be sources of 24-ipc in addition to (or in symbiosis with) algae and bacteria. This study is of high quality and great value to a wide range of audiences including biochemists and geobiologists. I do not have any major criticisms. I have only one comment regarding the text: Some paragraphs are hard to read because the discussed taxonomy covers two domains and many phyla and other taxonomic

levels. It is sometimes not always clear whether a specific taxon is a sponge, bacteria, or algae, so it would make sense to simplify or clarify the nomenclature wherever possible, such as by writing “pelagophyte algae” instead of “pelagophytes). Some additional suggestions are included below. The same arguably applies to the trivial names of sterols, which could either be fully explained when first mentioned or otherwise described.

We thank the reviewer for the thoughtful comments. We have added descriptors for the text and figures and hope the text and figures are much clearer now.

Table 1: Can the presence/absence of other sterol biosynthesis genes such as squalene monooxygenase, oxidosqualene cyclase, the SMTs and others be included in the table to further support the presence of sterol biosynthesis pathways in these organisms?

We have bolded the SMTs that are present in sterol biosynthesis gene clusters and denoted that in the legend. We have also expanded Figure 2 to include more gene clusters, and adjusted Figure 2 to denote the number of methylations performed by each SMT. We have also provided a supplementary file of the other sterol biosynthesis homologs we identified (Supplementary File 1).

Figure 1: Switch order of compounds V and IV in panel e?

We have chosen to keep the order to reflect the number of methylations rather than elution order.

Reviewer #3

Dear authors,

Sterol methyltransferases (SMTs) had been previously identified in sponge genomes. Brown et al. for the first time identify SMTs in bacteria as well (sponge-associated bacteria and free-living freshwater bacteria). They expressed these sponge and bacteria enzymes in *E. coli*, and tested their activity showing that these SMTs are functional: not only can they add a methyl group on carbon 24 of the side-chain but the same enzyme can sequentially build an isopropyl side-chain on carbon 24, thereby helping to produce C30 sterol precursors of C30 biomarker 24-ipc. The significance of these results is high in the field of paleo geochemistry and evolution, where the origin of co-occurring biomarkers 24-ipc and 26-mes in the Neoproterozoic is highly debated. These biomarkers may represent the first signs of sponge presence (i.e. the first animals) where sterol precursors are well known, or may be produced by other organisms. Brown et al. show for the first time that bacteria have the capacity to produce 24-ipc precursors, and that only one SMT is necessary for the isopropyl group. This study will also definitely significantly move the field of sterol biosynthesis forward, whether it be in sponges or in bacteria.

The main weakness of this study is that Brown et al. have not explored sponge species that actually produce high amounts of 24-ipc (or 26-mes) sterol precursors. All of the 8 sponge genomes investigated come from species which are not known to contain C30 sterols, so I was not surprised (as were the authors) their SMTs could not make C30 sterols. As for the metagenomes they come from sponge species with no C30 sterol precursors of 24-ipc/26-mes either. *A. aerophoba* does contain trace amounts of 24-ipc, but as many demosponges do (Zumberge et al., 2018. Table S5) and Brown et al. are aware of that (line 76). Altogether, sponge species from this manuscript are poor models to study the production of sponge biomarker precursors and eventually test the sponge biomarker hypothesis. Although it is clear the authors are aware of this weakness of the paper, it is only mentioned in the discussion (lines 163-165) although I think it should appear earlier in the main text. The authors need to explain from the start how the species sampled will limit their interpretations and conclusions on the “sponge biomarkers” per se. Also, the authors could add on Table 1 the known natural sterols of the species/genera investigated.

We thank the reviewer for their thoughtful comments. As pointed out by the reviewer, we are aware of the sampling issues and have made efforts to emphasize this in the “Sponge SMTs are functional” section of the manuscript. We also attempted to edit Table 1 as suggested but we could not find an elegant way to do this given differing types of sterol data we have for these sponges, and that not all of the SMTs in the table are associated with sponges. But we have added a supplementary table with more information on the sterols in these sponges (Supplementary Table 1).

Since *A. aerophoba* only has trace amounts of 24-ipc precursors, the bacteria with the sterol gene cluster in *A. aerophoba* discovered by Brown et al. may therefore produce very little amounts of 24-ipc precursors, in disagreement with the amounts observed in Neoproterozoic rocks. So, yes, some bacteria may contribute to 24-isopropyl sterol biosynthesis in demosponges, or in the environment, but as for the Rhizaria (Nettersheim et al., 2019; Love et al., 2020) this contribution does not seem to be significant. If Brown et al. agree with this reasoning, I think it should be made clearer in their discussion, and in the title. Maybe the title of the ms could be tuned down; ‘confounds’ is a rather strong word in my mind, suggesting that the authors will present overturn some of the eukaryotic biomarkers interpretations. But the selection of species prevents a real challenge of the sponge biomarker hypothesis.

We agree and have clarified our position in the discussion. We have changed the title to “Sterol methyltransferases in uncultured bacteria complicate eukaryotic biomarker interpretations” and hope this is sufficient.

Here are a few other more specific remarks:

Lines 57-59. This sentence suggests that the species selected were good species to get insights into sterol 24-ipc precursors, but it is not that simple, as stated previously. The authors need to acknowledge at this point already the limits of using these sponge species.

We added the following text to make the limits of the sponge species we used in this study clearer:

Given the current lack of publicly available sequencing data from sponges known to contain 24-propyl sterols, the function of SMTs from these species could not be verified in this study. We therefore chose eight SMT homologs identified in publicly available genomes and transcriptomes from eight sponges of the Demospongiae, Homoscleromorpha, and Calcarea classes, each of which have been shown to contain 24-ethyl sterols at the species or genus level (Supplementary Table 1).

Line 61-65. If I understood correctly, the sponge SMT was only shown to produce one alkylation at a time. How come? Since, SMT is producing 24-methylenecholesterol from desmosterol, why can't it continue to produce 24-ethyl sterols in one go (as the bacterial SMTs do)? I am missing in Fig 1c a chromatogram of an assay with a sponge SMT. Could the authors make this clearer in the manuscript?

We understand the confusion. The sponge (*Chondrilla nucula*) SMT included in Figure 1 also methylated desmosterol, and we have added this chromatogram to Figure 1c. We have also added text to the results section to clarify that each SMT we tested that could produce 24-ethyl sterols could also produce 24-methyl sterols from desmosterol.

Line 76. The authors should probably cite Zumberge et al., 2018, Table S5, instead of Zumberge's PhD thesis, or both.

We changed this citation to the 2018 paper.

Line 88. It is unclear how the 14 bacterial SMT were selected for heterologous expression and SMT assay. Can the authors explain this in the text? How many SMT were identified in the first place, are they that common? Did you notice if these SMTs were always found in sterol bacterial gene clusters? Since these are the first bacterial SMTs to be discovered I would give a bit more details about these.

We added text to the bioinformatics analysis section of the methods describing our selection of SMTs. Metagenomic SMTs are not always found in sterol gene clusters and we are not sure how many are bacterial. We don't have numbers on this but are currently undertaking a broader analysis of this as part of a follow-up study.

Line 96 and 111, the authors mean '14' not '24', right?

We mean 24. 10 from sponge metagenomes and 14 from other metagenomes.

Line 159. Here the authors must specify that the sterols of *Oscarella carmela* and *Sycon ciliatum* have not been analysed so the "previous hypotheses" mentioned are based on the possession of a single SMT copy (Gold et al., 2016). Maybe these two sponge species only naturally produce 24-methyl sterols, as the assay suggests. But yes, I

agree that depending on the species, sponge SMTs seem to add one or several methyl groups on carbon 24.

We added text stating that sterols from *O. carmela* and *S. ciliatum* haven't been reported. We also clarified the "previous hypothesis" that all sponges encode bifunctional SMTs. We also added additional text on the work of Gold et al., 2016 to the introduction to help clarify this concept.

Line 171-172. Although a valid hypothesis, I do not see how your data can suggest that sponges and bacteria "may therefore be required to produce 24-isopropyl sterols" together. Especially since you have not studied a STM from a sponge with high propylated sterols.

We have expanded this section in hopes of clarifying this hypothesis.

Line 200-205. Suggested improvement: have the authors tried to detect Chlamydiae in the metagenome of *Aplysina aerophoba*?

I would like to point this recent preprint claiming to have found substantial diversity of Chlamydiae in sponges (*Halichondria* and *Haliclona* species). They also identified sterol reductases that they suggest could have been acquired by HGT from other bacteria or from eukaryotes.

<https://www.biorxiv.org/content/10.1101/2021.12.21.473556v1.abstract>

We have not tried to detect Chlamydiae in the metagenome of *Aplysina aerophoba* but it looks like Chlamydiae OTUs were found in *A. aerophoba* in Thomas et al., 2016, Nat Comms. We have removed the discussion of Chlamydiae as a potential source of freshwater 24-ipc at the suggestion of Reviewer 1 and instead discussed Chlamydiae in sponges based on this suggestion.

Line 356. Give web link to the JGI IMG.

Link provided.

Line 357. I could not access the website www.compagen.org. Maybe I was unlucky, but please double-check that the link is working.

This website was down for some time, but it looks like it was moved recently. We have provided the new link.

Line 403. It is not clear in the text if replicates of these assays were done? at least for some of the SMTs.

All SMTs were tested at least twice with each substrate and produced the same products each time, we have added text to reflect this in the methods section.

REVIEWERS' COMMENTS

Reviewer #1 (Remarks to the Author):

The authors have answered each point raised by R1 and R3 comprehensively and they have made a good effort to incorporate edits that help clarify the take home messages from their study. As such, I have no further comments and/or queries and I recommend publication of this revised version of the ms.

Reviewer #3 (Remarks to the Author):

I am satisfied with the replies and the changes/rewording made by the authors. I have no further comments.

Reviewer #1 (Remarks to the Author):

The authors have answered each point raised by R1 and R3 comprehensively and they have made a good effort to incorporate edits that help clarify the take home messages from their study. As such, I have no further comments and/or queries and I recommend publication of this revised version of the ms.

Thank you.

Reviewer #3 (Remarks to the Author):

I am satisfied with the replies and the changes/rewording made by the authors. I have no further comments.

Thank you.